# Simultaneous and sensitive quantification of protein and low molecular weight persulfides, polysulfides and H$_2$S in biological samples

Jan Lj. Miljkovic[1], Nils Burger[1], Chak Shun Yu [1], Alexander H. Harkiss[2], Stefan Warrington [2], Stuart T. Caldwell [2], Scott A. Jones[1], Jordan J. Lee[3], Dunja Aksentijevic [4], Andrew M. James [1], Thomas Krieg [3], Richard C. Hartley [2] ✉ & Michael P. Murphy [1,3] ✉

H$_2$S reversibly modifies low molecular weight (L$_{MW}$SH) and protein (PrSH) thiols to form persulfides (RSS$^-$) and polysulfides (RS(S)$_n$S$^-$) for antioxidant defence and regulation of activity. However, our understanding of the biological significance of these processes is hampered by our inability to quantify these modifications. We develop a sensitive LC-MS/MS procedure that traps the sulfur atom of H$_2$S, and the terminal sulfur atom of RSS$^-$ and RS(S)$_n$S$^-$ as diagnostic products in biological samples. In parallel, we also trap internal S atoms of RS(S)$_n$S$^-$, enabling quantification of H$_2$S, RSS$^-$ and RS(S)$_n$S$^-$. L$_{MW}$S(S)$_n$S$^-$ and PrS(S)$_n$S$^-$ are determined simultaneously in the same sample. Glutathione (GSH) is the most abundant L$_{MW}$SH so we develop an orthogonal approach to quantify GSS$^-$, enabling corroboration of L$_{MW}$SS$^-$ measurements by sulfur atom trapping. We demonstrate in systems from proteins to ex vivo tissues how these approaches enable exploration of persulfidation in biological systems.

Long regarded as a toxic environmental pollutant and metabolic byproduct[1], hydrogen sulfide (H$_2$S) is now recognised, alongside nitric oxide (NO) and carbon monoxide (CO), as an important bioactive molecule[2,3]. Hydrogen sulfide (H$_2$S) is predominantly produced enzymatically through the activity of cystathionine β-synthase (CBS), cystathionine γ-lyase (CTH), 3-mercaptopyruvate sulfurtransferase (MPST) and the methanethiol oxidase selenium-binding protein 1 (SELENBP1)[3–6]. Furthermore, gastrointestinal microbiota also contribute to systemic H$_2$S levels[7,8].

H$_2$S is catabolised by the coupled activity of the mitochondrial enzymes sulfide:quinone oxidoreductase (SQOR), persulfide dioxygenase (ETHE1), thiosulfate sulfur transferase (TST; Rhodanese) and sulfite oxidase (SO) to generate thiosulfate (S$_2$O$_3^{2-}$), sulfite (SO$_3^{2-}$) and sulfate (SO$_4^{2-}$)[9]. At physiological pH, H$_2$S (pKa ~7) exists predominantly (~70%) as the hydrosulfide anion (HS$^-$) while the conjugate acid freely diffuses through biological membranes to access the catabolic machinery within mitochondria[4]. This balance between H$_2$S production and catabolism can be altered through changing enzyme activity or expression levels by altering the diet[10], or by modulating mitochondrial H$_2$S degradation[4,11]. For example, H$_2$S accumulates during ischaemia when reduction of the mitochondrial Coenzyme Q (CoQ) pool prevents H$_2$S consumption by SQOR[12,13].

Changes in H$_2$S levels are potential physiological signals; however, unlike NO, H$_2$S does not produce secondary messengers[4]. H$_2$S can only

[1]Medical Research Council-Mitochondrial Biology Unit, University of Cambridge, Cambridge, UK. [2]School of Chemistry, University of Glasgow, Glasgow, UK. [3]Department of Medicine, University of Cambridge, Cambridge, UK. [4]William Harvey Research Institute, Bart's and the London Faculty of Medicine and Dentistry, Queen Mary University of London, London, UK. ✉e-mail: Richard.Hartley@glasgow.ac.uk; mpm37@cam.ac.uk

act directly, for example, by binding to metal centres in proteins such as cytochrome oxidase[14], or by reacting with NO to modulate its activity[15]. H$_2$S can also act indirectly through the persulfidation of low molecular thiols (L$_{MW}$SH) such as glutathione (GSH) or of protein cysteine residues (PrSH)[3,4]. Persulfide formation occurs by the reaction of H$_2$S with a thiol (RSH) that has been oxidised to a sulfenic acid, S-nitrosothiol, or mixed disulfide[3,4,16]. Alternatively, a low molecular persulfide (L$_{MW}$SS$^-$) or a protein persulfide (PrSS$^-$) can transpersulfidate a PrSH[3,4,17,18]. In addition to persulfides (RSS$^-$), polysulfides (RS(S)$_n$S$^-$) have also been reported to occur in biological systems and could arise from the reaction of H$_2$S with derivatives of existing persulfides[19,20]. However, the modes of formation and existence of polysulfides in vivo are currently unclear.

Protein persulfides (PrSS$^-$) are readily reversed by the thioredoxin (Trx) and GSH/glutaredoxin (Grx) protein thiol reduction systems[5,21,22]. Reversible formation of PrSS has been proposed as a posttranslational modification (PTM) that changes the function and localisation of proteins[2,4,22]. In addition, the reaction of H$_2$S with reversibly oxidised protein thiols to form a PrSS$^-$, followed by reduction back to a PrSH by the Trx/GSH/Grx systems, prevents irreversible loss of thiols through oxidation to a sulfinic or sulfonic acid[21,22]. Furthermore, persulfides themselves are more potent antioxidants than thiols[23]. However, considerable uncertainty remains about the (patho)physiological roles of thiol per- or polysulfidation.

Much of the current evidence for protein persulfidation comes from semi-quantitative and qualitative chemoselective labelling approaches to identify the cysteine residue modified on the protein[24–27]. A major constraint on determining the roles and importance of H$_2$S, RSS$^-$ and RS(S)$_n$S$^-$ is the difficulty in quantifying their absolute levels in biological systems. This is due to the limitations in accuracy, precision and sensitivity, as well as uncertainty over the nature of the species being detected by current methods[22,28,29]. While there are many indirect methods that respond to changes in levels of reactive sulfur, these are prone to artifacts due to the lack of selectivity for the reactive sulfur species[28,30]. Thus, the most promising quantitative approaches are those that rely on trapping the reactive sulfur atom(s) in a diagnostic product and then quantifying these, typically by mass spectrometry (MS)[28,31–33]. An initial approach to the quantification of total tissue polysulfides was to use a phosphine that incorporates the sulfane sulfur atom into a diagnostic derivative, in conjunction with isotope dilution and liquid chromatography tandem mass spectrometry (LC-MS/MS)[31,32]. This led on to methods that require alkylation of the reactive sulfur atom as the trapping reaction[28]. A number of potential alkylating agents, such as N-ethyl maleimide, led to loss of reactive sulfur, and thus the field has focused on iodoacetyl[28] or monobromobimane derivatives[28,34,35], although some concerns remain about the inadvertent loss of the reactive sulfur[36]. Among these alkylating agents, most work has been done using β-(4-hydroxyphenyl) ethyl iodoacetamide (HPE-IAM)[37] with related methods using N-iodoacetyl L-tyrosine methyl ester iodoacetamide (TME-IAM)[33]. These approaches have been used to trap reactive sulfur directly[33,37]. Related methods have assessed "total" H$_2$S after its release by reduction of biological samples, followed by trapping with ethyliodoacetate[38] or monobromobimane[34,35].

There are a number of limitations to the above approaches when quantifying reactive sulfur in biological systems: they only partially distinguish between H$_2$S, RSS$^-$ and RS(S)$_n$S$^-$; the simultaneous detection of H$_2$S, RSS$^-$ and RS(S)$_n$S$^-$ is difficult; assessment of L$_{MW}$SS$^-$ and PrSS$^-$, or their corresponding polysulfides is challenging; the sensitivity of detection is low; the synthesis of stable isotope-labelled internal standards that is required for absolute quantification is not done routinely; and in many cases the slow processing of samples ex vivo renders uncertain the trapped reactive sulfur species[38].

To address all of these issues we have developed a sensitive LC-MS/MS method for the simultaneous and quantitative measurement of H$_2$S, L$_{MW}$SS$^-$ and PrSS$^-$, as well as their corresponding polysulfides, in a range of biological samples. Here, we describe the development of this strategy and apply it to systems ranging from isolated proteins to ex vivo tissues.

## Results

### Trapping H$_2$S, RSS$^-$ and RS(S)$_n$S$^-$ as stable products

To quantify H$_2$S, RSS$^-$ and RS(S)$_n$S$^-$ within samples we developed a sulfur atom trapping approach that incorporates the target S atom into diagnostic products that can be sensitively quantified by LC-MS/MS. Initially, we explored the use of derivatised phosphines due to their reaction with RSS$^-$ to incorporate the sulfane sulfur atom[32]. However, we found that the reactivity of phosphines with oxygen to form a phosphine oxide, along with the relatively long reaction times required and the instability of phosphine sulfides, decreased sensitivity. Instead, we utilised the high nucleophilic reactivity of RSS$^-$ and RS(S)$_n$S$^-$ (due to their low pK$_a$ and the α-effect of the non-terminal sulfur atom) that enables reaction with iodoacetamide (IAM) to generate reaction products that incorporate the terminal sulfane sulfur (Fig. 1)[16,39]. To enhance the sensitivity of detection of the diagnostic products, instead of IAM we used a triphenylphosphonium cation (TPP$^+$) derivative of iodoacetamide (IAM-TPP)[40]. Incorporation of the fixed positive charge of the TPP cation greatly enhances the sensitivity of detection of the products by LC-MS/MS, as we have used extensively in other approaches to quantify evanescent reactive species in biological systems[41,42].

In procedure A, RSS$^-$ and RS(S)$_n$S$^-$ were reacted with IAM-TPP[40] to carbamidomethyl-TPP thioethers (RSS-CAM-TPP and RS(S)$_n$S-CAM-TPP) while simultaneously blocking free thiols (RS-CAM-TPP) (Fig. 1). This trapping procedure will also convert H$_2$S into S(CAM-TPP)$_2$. Reduction with TCEP, followed by the addition of excess iodoacetamide (IAM), releases a stoichiometric amount of CAM-S-CAM-TPP from RSS-CAM-TPP and RS(S)$_n$S-CAM-TPP that incorporates the terminal sulfur atom from RSS$^-$ and RS(S)$_n$S$^-$. Thus, procedure A generates two products tagged with TPP for MS detection: CAM-S-CAM-TPP from the *terminal* S atom in RSS$^-$ and RS(S)$_n$S$^-$ and S(CAM-TPP)$_2$ from H$_2$S (Fig. 1). It also releases S(CAM)$_2$ that is not tagged for MS detection.

In addition to RSS$^-$, there is also the possibility of polysulfide (RS(S)$_n$S$^-$) formation, but their amounts and significance in biological systems remain unclear[43]. The method in procedure A quantifies the *terminal* S atom in both RSS$^-$ and RS(S)$_n$S. Therefore, we also developed procedure B to quantify in parallel *internal* sulfane sulfur atoms, RS(S)$_n$S$^-$. This was done by first alkylating RSH, RSS$^-$ and RS(S)$_n$S$^-$ with IAM (Fig. 1). Subsequent reduction with TCEP alongside trapping with excess IAM-TPP generates CAM-S-CAM-TPP from the *terminal* sulfur atom of RS(S)$_n$S$^-$ and S(CAM-TPP)$_2$ from all the *internal* sulfane sulfur atoms RS(S)$_n$S$^-$ (Fig. 1, Supplementary Fig. 1). Thus, by applying procedures A and B we can generate TPP products diagnostic of H$_2$S, *terminal* S atom in RSS$^-$ and RS(S)$_n$S$^-$ and *internal* S atoms in RS(S)$_n$S$^-$.

Another way of considering the chemical basis of these complementary approaches is to note that in procedure A, we leverage the fact that in solution, the deprotonated terminal perthiolate and perthiolate-like sulfur atoms in polysulfides have *nucleophilic* chemical properties and thus react with IAM/IAM-TPP. In contrast, the internal sulfane sulfurs of polysulfides are quite different chemically from the terminal sulfur atoms and thus react initially as electrophiles.

The next step was to establish LC-MS/MS quantification of CAM-S-CAM-TPP and S(CAM-TPP)$_2$. We generated standards, along with deuterated derivatives (Supplementary Fig. 2). Product scans of the precursors gave characteristic product ion masses, which were attributed to fragmentation of the TPP-moiety of the compounds, enabling detection by LC-MS/MS (Fig. 2a). Linear standard curves enabled detection of both compounds sensitively in the low fmol range (Fig. 2b).

Together, these two procedures enable quantification of the total H$_2$S, terminal S atoms in RSS$^-$ and both terminal and internal S atoms in

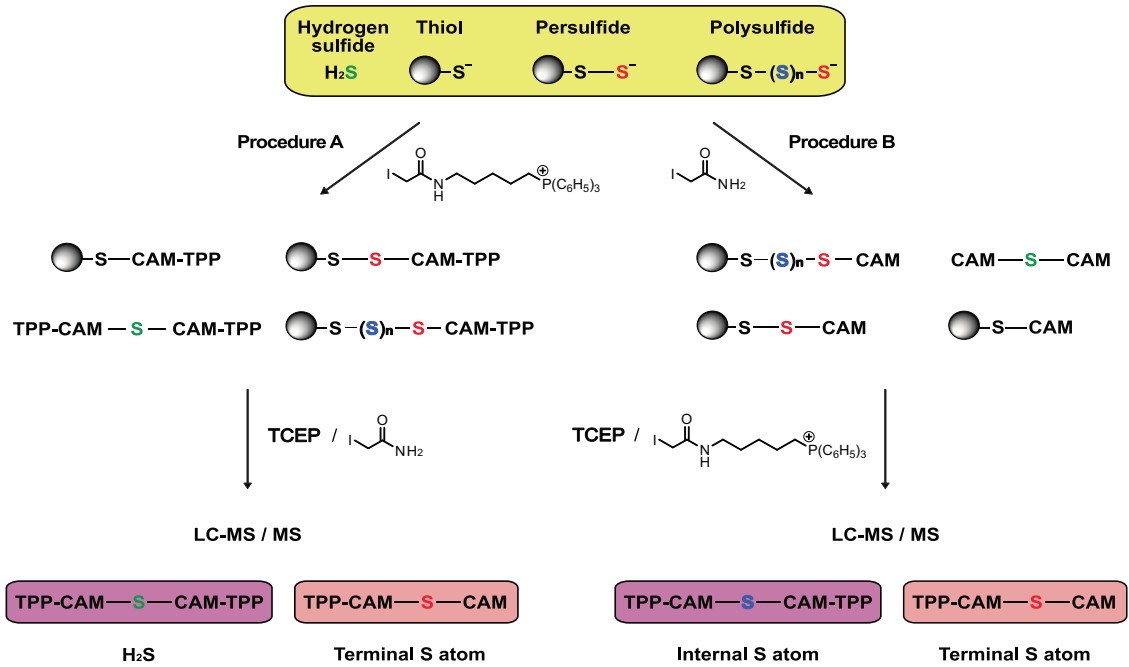

**Fig. 1 | Sequential alkylation, reduction, and further alkylation of H2S, RSS⁻ and R(S)ₙS⁻.** To quantify H2S, RSS⁻ and R(S)ₙS⁻, samples were first reacted with excess IAM-TPP (Procedure A) or IAM (Procedure B). H2S present will be converted to the readily detectable S(CAM-TPP)₂ in Procedure A and to S(CAM)₂ in Procedure B, which is far less readily detected due to its lack of charge. The CAM-blocked persulfide and polysulfide products were then reduced with TCEP, followed by treatment with excess IAM (Procedure A) or IAM-TPP (Procedure B). This generates CAM-S-CAM-TPP from terminal S atoms in persulfides and polysulfides (Procedure A), and S(CAM-TPP)₂ from internal sulfane S atoms (Procedure B). These products can then be extracted and quantified by LC-MS/MS enabling simultaneous measurement of H2S, RSS⁻ and RS(S)ₙS⁻ in biological samples.

RS(S)ₙS⁻. They can also be readily adapted to quantify both $L_{MW}SS^-$/$L_{MW}S(S)_nS^-$ and PrSS⁻/PrS(S)ₙS⁻, as will be demonstrated below.

## Quantification of persulfides and polysulfides on PrSH

We first used procedures A and B shown in Fig. 1 to quantify PrSS⁻ and PrS(S)ₙS⁻ on purified proteins. To do this, we used a modified human thioredoxin 1 (Trx1) incorporating a streptavidin binding protein (SBP)-tag and containing only a single Cys residue[36] (Supplementary Fig. 3a–c). The SBP-tag enabled binding to NeutrAvidin beads, which facilitated treatment with Na2S2, followed by washing and processing without precipitation and resuspension of the protein (Supplementary Fig. 3d), which makes an ideal model system. Reaction of the exposed thiol with Na2S2 should lead to both persulfide and polysulfide formation (Fig. 3a). Treatment with Na2S2 followed by analysis using procedure A led to the concentration-dependent and saturable generation of CAM-S-CAM-TPP, consistent with detection of the terminal S atoms generated by Na2S2 forming PrSS⁻ and PrS(S)ₙS⁻ on Trx1 (Fig. 3b). Na2S2 treatment followed by analysis using procedure B also led to a dose-dependent formation of S(CAM-TPP)₂ consistent with PrS(S)ₙS⁻ generation (Fig. 3c). Interestingly, S(CAM-TPP)₂ formation assessed by procedure B did not saturate, consistent with the reaction of initially formed PrSS⁻ or PrS(S)ₙS⁻ with Na2S2 to form further PrS(S)ₙS⁻. One possible caveat with procedure B is that in the absence of IAM/IAM-TPP as a capping agent, extended exposure to TCEP could sequester internal sulfane sulfur atoms as TCEP = S[44] through nucleophilic attack on the internal sulfur atom (Supplementary Fig. 4)[45]. If this occured, it could under-report the levels of polysulfide sulfane sulfur, thus in our experiments we minimised incubation with TCEP without capping agent. No CAM-S-CAM-TPP or S(CAM-TPP)₂ was formed without Trx1 or Na2S2 (Fig. 3b, c). Treatment of bound Trx1 with DTNB followed by Na2S should generate only PrSS⁻[16] (Fig. 3a). This led to the detection of CAM-S-CAM-TPP by procedure A (Fig. 3d), and this was prevented by the presence of ZnCl2 which sequesters H2S preventing the formation of persulfides[46] (Fig. 3d) while S(CAM-TPP)₂ was not detectable by procedure B. Na2S alone did not generate CAM-S-CAM-TPP (Fig. 3d).

To extend this approach beyond protein bound to beads, we next analysed human recombinant Cofilin 1 treated with MitoPerSulf, which forms persulfides (PrSS⁻) on Cys39 and Cys139, as was demonstrated previously by LC-MS peptide profiling[47]. Treatment of Cofilin 1 with MitoPerSulf was followed by procedure A using precipitation and resuspension to remove the alkylating agent IAM-TPP. The alkylated protein was then treated with TCEP/IAM, precipitated again, and the supernatant analysed by LC-MS/MS. Treatment of cofilin 1 with Mito-PerSulf generated CAM-S-CAM-TPP when analysed by procedure A (Fig. 3e), consistent with PrSS⁻ formation on Cofilin 1[26].

We next extended the analysis of single proteins to a cell lysate prepared by freeze-thawing and sonicating human embryonic kidney cells (HEK 293). Lysates were treated with Na2S2 or Na2S, followed by alkylation with IAM-TPP or IAM (procedures A or B, respectively), and then adsorbed onto mixed hydrophobic/hydrophilic solid matrix SP3 beads to facilitate processing of the protein sample without precipitation and resuspension[48,49] (Supplementary Fig. 5a). The bead-adsorbed proteins were then processed by procedure A to generate CAM-S-CAM-TPP (Fig. 3f) and by procedure B to generate S(CAM-TPP)₂ (Fig. 3g). Treatment with Na2S2 led to the dose-dependent formation of PrSS⁻ and PrS(S)ₙS⁻ that was reversed with TCEP (Fig. 3f, g) and there was no PrSS⁻ background detected in the untreated cell lysate.

Cell lysate samples on SP3 beads were treated with Na2S2 to introduce persulfides, or with Na2S as a control, labelled with IAM-TPP, reduced with TCEP, and then further alkylated with IAM. After elution from the SP3 beads and separation by SDS-PAGE the blots were immunoprobed with anti-TPP rabbit antiserum[50] to generate the overall protein-bound TPP signal, relative to vinculin as a loading control. The TPP signal was weaker in samples treated with Na2S2

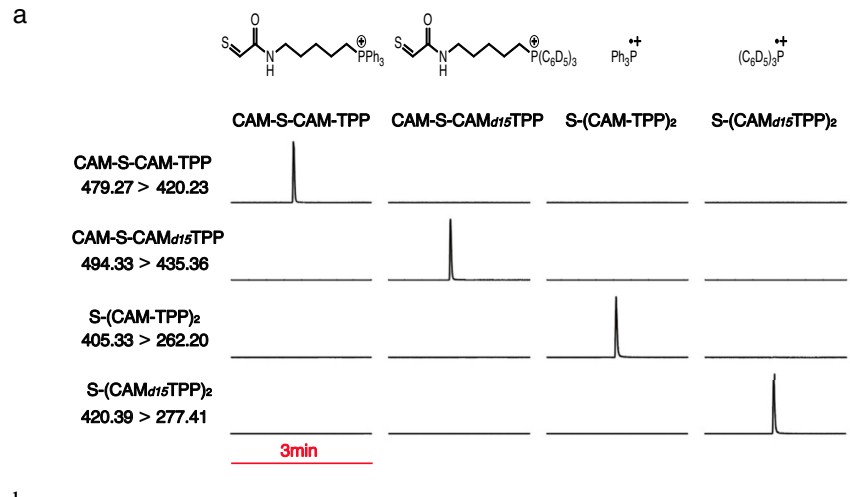

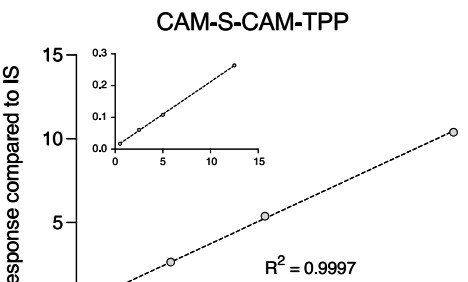

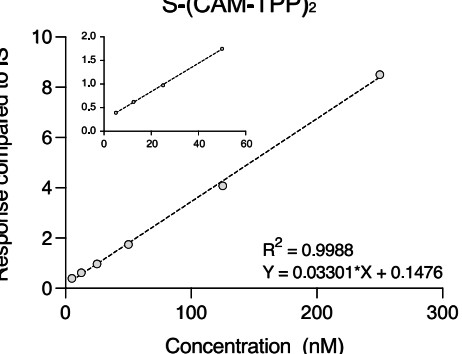

**Fig. 2 | LC-MS/MS quantification of CAM-S-CAM-TPP and S(CAM-TPP)₂.**
**a** Representative multiple reaction monitoring (MRM) chromatograms from LC-MS/MS analyses of 0.5 pmol compound displaying the simultaneously measured m/z transitions diagnostic for CAM-S-CAM-TPP, S(CAM-TPP)₂ or their respective deuterated internal standards. All chromatograms are normalised to the highest peak for each sample. **b** Standard curves for the detection of CAM-S-CAM-TPP and S(CAM-TPP)₂ by LC-MS/MS. The MS response at different compound concentrations was normalised to that of 0.5 pmol of the corresponding deuterated internal standard. Source data are provided as a Source Data file.

rather than Na₂S (Supplementary Fig. 5b). This is consistent with TPP-IAM-based alkylation of PrSH as well as of PrSS⁻ and PrS(S)$_n$S⁻ terminal sulfur atoms. The TPP labelling of PrSH is stable, while that of PrSS⁻ and PrS(S)$_n$S⁻ is via a disulfide and is consequently lost upon reduction and further alkylation with IAM, in agreement with Fig. 1. Treatment of mouse heart homogenates generated under native conditions with Na₂S₂ also led to the formation of PrSS⁻ and PrS(S)$_n$S⁻ detected by the same procedures (Supplementary Fig. 5c, d).

To assess the proportion of available protein thiols modified by Na₂S₂, we quantified the thiols and protein immobilised on the SP3 beads (Supplementary Fig. 5e). This showed that ~430 pmol of thiols were bound to beads, which corresponds to ~15 pmol of PrSS⁻ and PrS(S)$_n$S⁻, or per-/polysulfidation of 3.5% of available protein thiols under these conditions. Thus, this sulfur atom trapping LC-MS/MS approach can be used to quantify PrSS⁻/PrS(S)$_n$S⁻ on isolated proteins and in crude protein mixtures.

**Quantification of persulfides and polysulfides on L$_{MW}$SH**
We next extended these procedures to quantify L$_{MW}$SS⁻. First, we generated L$_{MW}$SS⁻ by incubating GSH with DTNB to generate GS-TNB, which was then converted into GSS⁻ by addition of Na₂S, as indicated by TNB release at both steps (Fig. 4a)[16]. We then reacted GSH with 1 equivalent (eqv.) of DTNB followed by 1 eqv. of Na₂S and then applied procedure A and quantified CAM-S-CAM-TPP (Fig. 4b). This showed

formation of CAM-S-CAM-TPP that was prevented by pretreatment with the thiol alkylator 4-chloro-7-nitrobenzofurazan (CNBF), or treatment with excess TCEP prior to addition of IAM-TPP (Fig. 4b). In contrast, when similar samples were processed by procedure B, negligible amounts of polysulfides (internal sulfane sulfur) were detected (Fig. 4c). We then reacted 10 μM GSH with 0 to 1 equiv. of DTNB followed by 10 μM Na₂S and then applied procedure A and quantified [CAM-S-CAM-TPP] (Fig. 4d). This showed that [CAM-S-CAM-TPP] increased with the amount of DTNB added, up to 1 eqv. of DTNB, indicating that this procedure responded to increasing L$_{MW}$SS⁻. Note that the yield of CAM-S-CAM-TPP was considerably lower than the amount of Na₂S added in all cases, due to the use of an open system and consequent loss of H₂S to the atmosphere. Absence of Na₂S prevented CAM-S-CAM-TPP formation, while excess Na₂S only slightly increased [CAM-S-CAM-TPP]. Using procedure B to assess RS(S)$_n$S⁻ under these conditions showed minimal S(CAM-TPP)₂ formation, except when there was excess Na₂S, consistent with its reaction with existing persulfides (Supplementary Fig. 6a). Incubating GSH with Na₂S₂ followed by procedure A showed some formation of [CAM-S-CAM-TPP] consistent with detection of the terminal sulfur of the polysulfides and any persulfide present (Fig. 4e). In addition, application of procedure B showed the generation of S(CAM-TPP)₂ (Fig. 4f), consistent with detection of the internal sulfane sulfur polysulfide formation. Thus, this approach can be used to monitor persulfides and

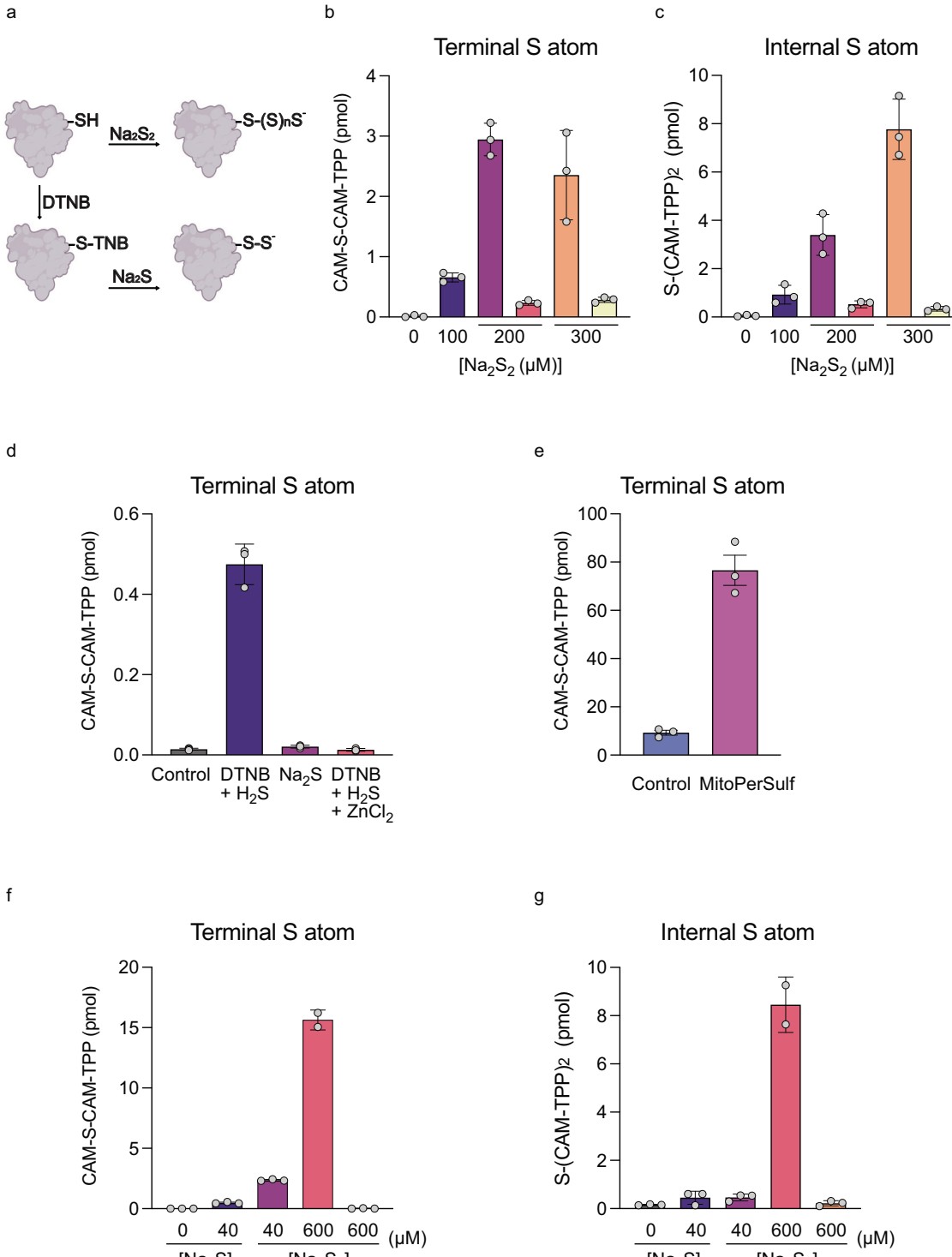

**Fig. 3 | Quantification of protein persulfides (PrSS⁻) and polysulfides (PrS(S)ₙS⁻).** **a** Schematic showing expected reactions of protein thiol with Na₂S₂, or with DTNB followed by Na₂S. **b**–**d** Reduced Trx1 (70 µg) was incubated with 0.5 mL Neutravidine-agarose bead slurry for 60 min, washed, treated with Na₂S₂ (0-300 µM) for 7 min (**b**, **c**) or with 20 µM DTNB for 20 min, followed by washing and exposure to Na₂S (15 µM) for 7 min (**d**). All samples were then washed and processed by Procedure A for the generation of CAM-S-CAM-TPP (**b**, **d**) or Procedure B for S(CAM-TPP)₂ (**c**). Indicated samples were treated with TCEP (5 mM) prior to analysis, or ZnCl₂ (1 mM) was added along with the Na₂S (**d**). Data are mean ± S.D. n = 3. **e** Human recombinant Cofilin 1 (11 µg) was incubated with 1 mM GSH ±

MitoPerSulf (100 µM) for 7 min, and then CAM-S-CAM-TPP products were quanti-fied by Procedure A. HEK 293 cell lysates (100 µg protein in 200 µL 50 mM HEPES buffer pH 7.4) were exposed to Na₂S (40 µM) or to Na₂S₂ (40-600 µM) for 7 min, alkylated with either IAM-TPP (**f**) or IAM (**g**), bound to the SP3 beads slurry and washed with HEPES buffer. Samples were then processed by procedure A (**f**) or procedure B (**g**). Data are mean ± S.D. n = 3 (n denotes the number of individual technical replicates). Selected protein structure elements in (**a**) were prepared with the assistance of ChemDraw software (version 23.1.2). Source data are provided as a Source Data file.

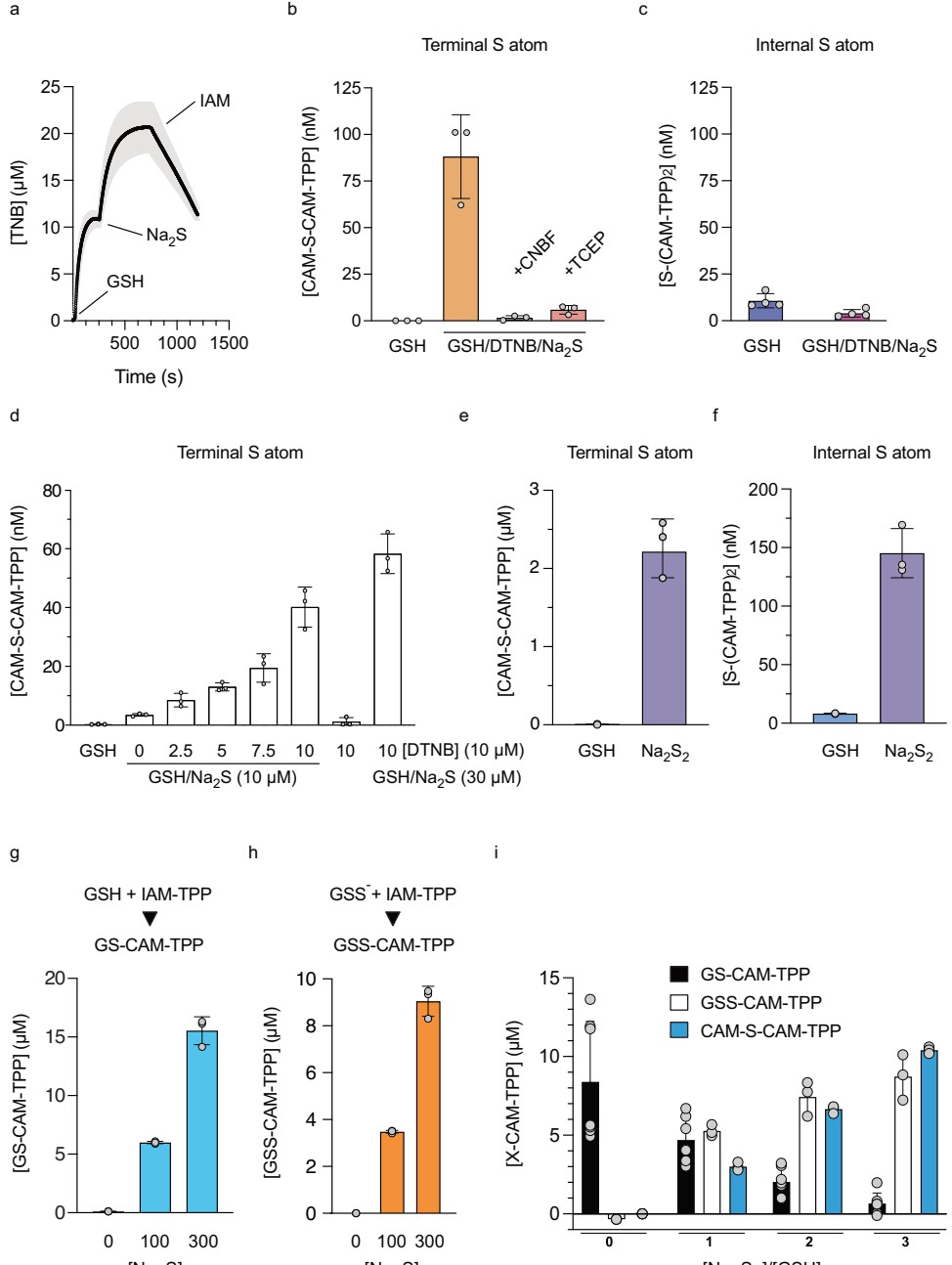

**Fig. 4 | Quantification of persulfides and polysulfides on GSH. a** To DTNB
(20 μM) were added GSH (20 μM), Na₂S (20 μM) and IAM (20 μM) and the produc-
tion of TNB monitored. SD shown by shading. **b** GSH (10 μM) was incubated ± DTNB
(10 μM) and then with Na₂S (10 μM). Samples were processed by Procedure A and
CAM-S-CAM-TPP quantified. Where indicated CNBF (10 μM) was added prior to
DTNB, or TCEP (100 μM) was added prior to IAM-TPP. **c** GSH incubated as in b ±
DTNB and then with Na₂S. Samples were processed by Procedure B and S(CAM-
TPP)₂ quantified. **d** GSH was incubated with DTNB (0 − 10 μM) and then with Na₂S
(0, 10, 30 μM). Samples were processed by Procedure A and CAM-S-CAM-TPP
quantified. **e, f** GSH (10 μM) was incubated with Na₂S₂ (0 − 30 μM). **e** samples were

processed by Procedure A and CAM-S-CAM-TPP quantified. **f** samples were pro-
cessed by Procedure B and S(CAM-TPP)₂ quantified. Reaction of Na₂S with GSSG.
GSSG (100 μM) was incubated with Na₂S (0, 100 or 300 μM), blocked with IAM-TPP
(1 mM) and GS-CAM-TPP (**g**) and GSS-CAM-TPP (**h**) quantified. **i** Reaction of GSH (10
μM) with Na₂S₂ (0, 10, 20 and 30 μM). Samples were then blocked with IAM-TPP and
GS-CAM-TPP and GSS-CAM-TPP quantified, or processed using Procedure A CAM-S-
CAM-TPP quantified. Data are mean ± S.D., n = 3 (**b**, **d**–**h**); n = 4, (**c**); n = 7 (GS-CAM-
TPP; **i**) n = 3 (GSS-CAM-TPP and CAM-S-CAM-TPP; **i**) where n denotes the number of
individual technical replicates. Source data are provided as a Source Data file.

polysulfides on L_MWSH. The amount of GSH converted to GSS⁻ ranged
from 0.4–20%, dependent on the conditions used.

To further assess L_MWSS⁻, we next set up a direct LC-MS/MS assay
for GSS⁻ itself. As GSH is the most abundant L_MWSH in vivo, GSS⁻ is
likely to be the dominant L_MWSS⁻ in a biological context. The basis of
this assay was to react GSS⁻ with IAM-TPP to generate GSS-CAM-TPP,

which can readily and sensitively be detected by LC-MS/MS due to its
positively charged TPP MS tag (Supplementary Fig. 7a). This enabled
the quantification of GSS⁻ against its corresponding internal deuter-
ated standard GSS-CAM-d₁₅TPP (Supplementary Fig. 7b). A similar
approach was also used to derivatise GSH with IAM-TPP to generate
GS-CAM-TPP (Supplementary Fig. 7a, b). Using these direct assays, we

showed that incubation of GSSG with $Na_2S$ followed by trapping with IAM-TPP led to the formation of GSH, detected as GS-CAM-TPP (Fig. 4g), and GSSH, detected as GSS-CAM-TPP (Fig. 4h). Our yield for the synthesis of $GSS^-$ from the reaction of GSSG and $H_2S$ was in agreement with a previous report[51]. Next, we generated GSSH by reaction of GSH with 0–3 eqvs. of $Na_2S_2$ followed by trapping with IAM-TPP (Fig. 4i). This showed a $Na_2S_2$ dose-dependent loss of GSH along with the accumulation of $GSS^-$ (Fig. 4i). In parallel, we also assessed these samples for the production of $L_{MW}SS^-$ using procedure A (Fig. 4i) which showed formation of CAM-S-CAM-TPP at a similar concentration to GSS-CAM-TPP (Fig. 4i). Under these conditions there was also some polysulfide formation as assessed by procedure B (Supplementary Fig. 6b) and $S(CAM-TPP)_2$ formation by procedure A consistent with $H_2S$ formation (Supplementary Fig. 6c).

Terminal alkylation of persulfides under some conditions leads to the conversion of the persulfide to a thioether with the loss of the sulfane sulfur[36]. Even though this rearrangement occurs far less with bulky alkylating agents that are similar in size to IAM-TPP[36], any conversion of the initial alkylated persulfide in procedure A to a thioether would artifactually decrease the amount of persulfide quantified (Supplementary Fig. 7c). Our generation of pure GSS-CAM-TPP enabled us to assess whether this impacted our approach by quantifying any conversion of GSS-CAM-TPP to its corresponding thioether GS-CAM-TPP in the presence of TCEP (Supplementary Fig. 7b). We incubated GSS-CAM-TPP with 0 to 10 equiv. of TCEP and assessed the production of GS-CAM-TPP (Supplementary Fig. 7d, e). Incubation of GSS-CAM-TPP with TCEP for 30 min led to the loss of GSS-CAM-TPP in proportion to the amount of TCEP added, but without formation of GS-CAM-TPP (Supplementary Fig. 7d, e). The detection of residual amounts of GS-CAM-TPP may have been due to contamination of the GSS-CAM-TPP, but this did not change in the presence of TCEP and was below the reliable limit of detection for GS-CAM-TPP. Hence, disulfide isomerisation followed by desulfuration to a thioether does not affect our quantification of persulfides. Thus, $L_{MW}SS^-$ can be determined by procedure A, and in parallel, the concentration of $GSS^-$, the probable dominant $L_{MW}SS^-$ in vivo, can be directly determined.

### Tissue protein and LMW persulfide levels

Next, we set out to apply these methods to assess steady-state levels of terminal S atoms due to $RSS^-/RS(S)_nS^-$ and $H_2S$ within tissues in vivo. To trap these evanescent species, we rapidly isolated and snap-froze a panel of mouse and rat tissues (Fig. 5a). Detection of the terminal S required only procedure A. The $H_2S$ present in the initial homogenate was assessed by detection of $S(CAM-TPP)_2$ after protein precipitation in the first aliquot. Treatment of the second aliquot with TCEP and IAM, followed by measurement of CAM-S-CAM-TPP after protein precipitation, gave the total amount of $RSS^-/RS(S)_nS^-$ present in the tissue homogenate, giving values that are consistent with earlier reports of overall tissue levels of persulfides using sulfane sulfur atom trapping[31,32]. Finally, in the third aliquot, protein precipitation prior to treatment of the supernatant with TCEP and IAM, followed by analysis of CAM-S-CAM-TPP, quantified total $L_{MW}SS^-/L_{MW}S(S)_nS^-$ present. For this, we used different standard curves from those used previously in order to avoid the matrix effects associated with analysis of a more complicated biological millieu (Supplementary Fig. 8). The levels of $H_2S$, total $RSS^-/RS(S)_nS^-$ and $L_{MW}SS^-/L_{MW}S(S)_nS^-$ are shown in Fig. 5b–d. Subtracting $L_{MW}SS^-/L_{MW}S(S)_nS^-$ from the total $RSS^-/RS(S)_nS^-$ in each sample gives $PrSS^-/PrS(S)_nS^-$ (Fig. 5e).

As the dominant $L_{MW}SH$ within tissues is GSH, comparison of $L_{MW}SS^-$ with $GSS^-$ detected as GSS-CAM-TPP levels should be informative. Initial assessments of the extract used to assess $H_2S$ levels (Fig. 5a) did not reliably quantify any GSS-CAM-TPP. This is likely in part due to the 50-fold higher limit of detection for GSS-CAM-TPP compared to CAM-S-CAM-TPP derived from $H_2S$, in conjunction with matrix effects from the tissue background. To overcome these issues,

we applied a Standard Addition Method (SAM)[52] to mouse liver samples, in which the tissue samples were derivatised with IAM-TPP to trap any $GSS^-$ as GSS-CAM-TPP, while also being spiked with a concentration range of GSS-CAM-TPP followed by quantification of total GSS-CAM-TPP. Extrapolation of the resulting plots gave an intercept on the x-axis that corresponds to the amount of GSS-CAM-TPP in the unspiked samples (Fig. 5f). From these experiments, we estimated the level of $GSS^-$ in mouse liver as $12.8 \pm 2.8$ pmol/mg wet weight (Fig. 5g), ~20% lower than the $L_{MW}SS^-$ determined by Procedure A in $16 \pm 1.6$ pmol/mg wet weight (Fig. 5d). Thus, the two measures are mutually consistent and show that $GSS^-$ is the dominant $L_{MW}SS^-$ within tissues in vivo.

### Persulfide and $H_2S$ status during ischaemia and oxidative stress

The relationship between levels of $H_2S$ and thiol persulfidation is not well understood. To see whether an elevation in $H_2S$ levels affected persulfidation, we investigated cardiac ischaemia, when the lack of oxygen and consequent reduction of the mitochondrial Coenzyme Q pool prevents the oxidation of $H_2S$ by mitochondrial SQOR and is thus thought to lead to the accumulation of $H_2S$[12,13].

First, we investigated mouse hearts in vivo, where the heart was freeze-clamped immediately following death to assess normoxic in vivo levels of $H_2S$ and persulfides. To mimic ischaemia, some hearts were removed and then incubated ex vivo at 37 °C for 30 min under ischaemic conditions, which are known to replicate the metabolic changes that occur during ischaemia in vivo[53]. Snap-freezing the tissues during normoxia or ischaemia was followed by processing with IAM-TPP, enabling quantification of $S(CAM-TPP)_2$ and CAM-S-CAM-TPP. This showed that the levels of $H_2S$ increased during ischaemia (Fig. 6a), but that total persulfides and polysulfides did not (Fig. 6b). Next, we used the isolated Langendorff mouse heart model to expose isolated hearts to normoxia followed by ischaemia and then reperfusion. Analysis of $S(CAM-TPP)_2$ and CAM-S-CAM-TPP again showed that the levels of $H_2S$ increased during ischaemia and decreased upon reperfusion (Fig. 6c). However, there was no corresponding change in total persulfides or polysulfides (Fig. 6d). These changes were not due to the limitations of the Langendorff mouse heart model as we found comparable results on analysing in vivo ischaemia in the heart using the left anterior artery, descending (LAD) occlusion model (Supplementary Fig. 9), and from in vivo ischaemia in the brain using the middle cerebral aretery occlusion (MCAO) model (Supplementary Fig. 9). Together these findings show that during ischaemia there is elevation in $H_2S$ but that this does not in itself lead to an increase in total persulfides.

Persulfides have been predicted to arise during oxidative stress by the reaction of $H_2S$ with sulfenic acids or disulfides (Fig. 6e)[16]. To assess this possibility, we incubated mitochondria with $H_2O_2$ to generate sulfenic acids and disulfides, in the presence or absence of $H_2S$, and then assessed total persulfides and polysulfides (Fig. 6f) and the levels of $GSS^-$ (Fig. 6g). The combination of oxidative stress caused by $H_2O_2$, along with $H_2S$ led to the formation of persulfides, detected as CAM-S-CAM-TPP or as GSS-CAM-TPP, while $H_2O_2$ or $H_2S$ alone did not (Fig. 6f, g). In the absence of $H_2O_2$ and $H_2S$ control mitochondria contained $1.9 \pm 0.2$ nmol GSH/mg protein as determined by measurement of GS-CAM-TPP. Under these conditions, exposure to $H_2O_2$ and $H_2S$ led to formation of $76 \pm 6$ pmol $GSS^-$/mg protein (Fig. 6f), thus about 4% of the GSH pool was persulfidated, with minimal protein persulfidation.

## Discussion

Our understanding of the roles of $H_2S$, $RSS^-$ and $RS(S)_nS^-$ in biology has been constrained by the difficulty of quantifying these species[22,24–29]. Here we have addressed this issue by developing a suite of LC-MS/MS methods that enable the sensitive and simultaneous quantification of $H_2S$, $RSS^-$, $RS(S)_nS^-$ and $GSS^-$. Furthermore, by appropriate separation of protein and $L_{MW}$ fractions, we can assess both protein and low

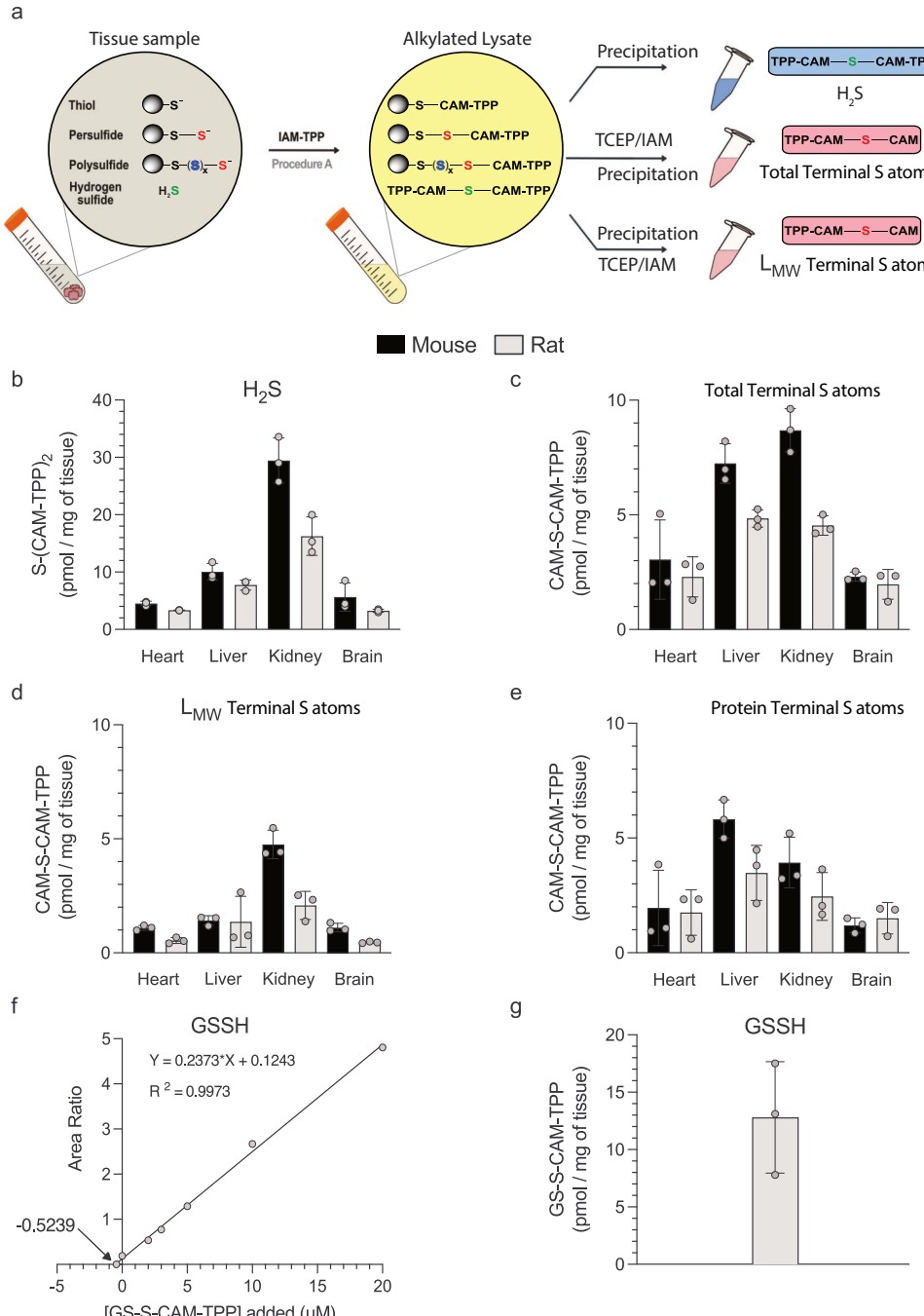

Fig. 5 | Steady-state H₂S and persulfides in rodent tissues in vivo. a Tissues were lysed in the presence of IAM-TPP, divided into three and processed as indicated. b H₂S detected as TPP-CAM-S-CAM-TPP. c Total persulfides detected as TPP-CAM-S-CAM. d LMW persulfides detected as TPP-CAM-S-CAM. e Protein persulfides quantified as (c, d). f Representative SAM analysis of mouse liver homogenate incubated with IAM-TPP, spiked with GSS-CAM-TPP and processed to quantify GSS-CAM-TPP. The x-axis intercept is the GSS-CAM-TPP concentration of the unspiked sample. g The experiment in (f) was repeated on two biological replicates and normalised to tissue weight to quantify GSS-CAM-TPP. Data are means ± SD, n = 3 (n denotes the number of individual biological replicates). Selected protein structure elements in (a) were prepared with the assistance of ChemDraw software (version 23.1.2). Source data are provided as a Source Data file.

molecular weight RSS⁻ and RS(S)ₙS⁻ simultaneously within a range of biological systems, from isolated proteins, to mitochondria and on to tissue homogenates. The parallel quantification of the persulfide of GSH, the dominant L_MWSH in biology, provides a further orthogonal corroboration of these measurements.

Our new suite of methods builds on earlier developments that trap the reactive sulfur atom(s) in a diagnostic product that is then quantified by mass spectrometry[28,31–35,37]. However, our approach is a significant advance on these earlier methods, for the following

reasons: snap-freezing of samples followed by reaction with IAM-TPP/IAM enables us to trap and stabilise all reactive sulfur species simultaneously through their nucleophilic reactions to generate stable products, rather than leaving labile moieties that may be modified during sample workup; incorporation of the fixed positive charge of the TPP cation into the diagnostic species greatly enhances sensitivity of detection by LC-MS/MS down to the low fmol level[41,42]; the simultaneous detection of H₂S, RSS⁻, RS(S)ₙS⁻ and GSS⁻ in biological samples is a major advance over previous approaches; and the synthesis of a

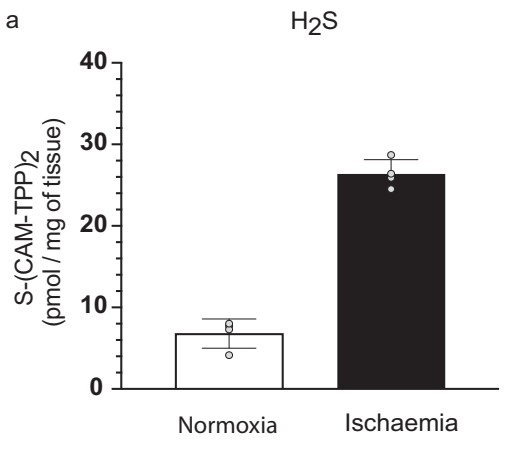

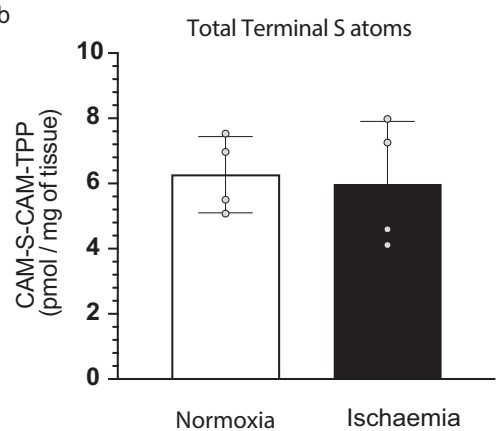

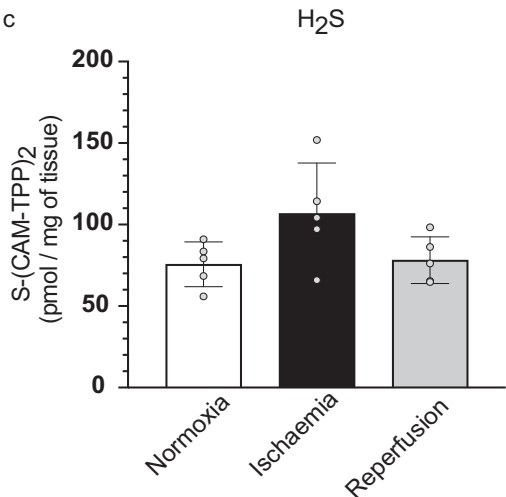

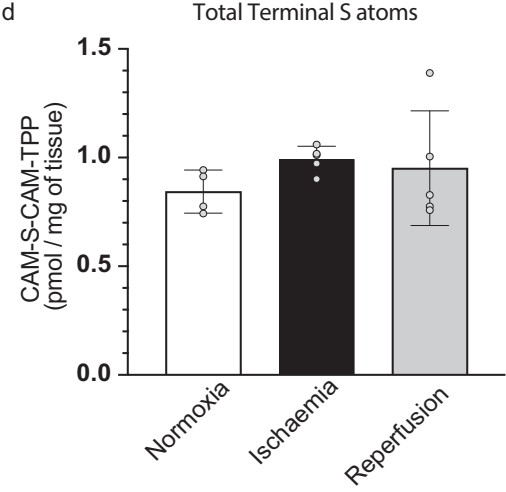

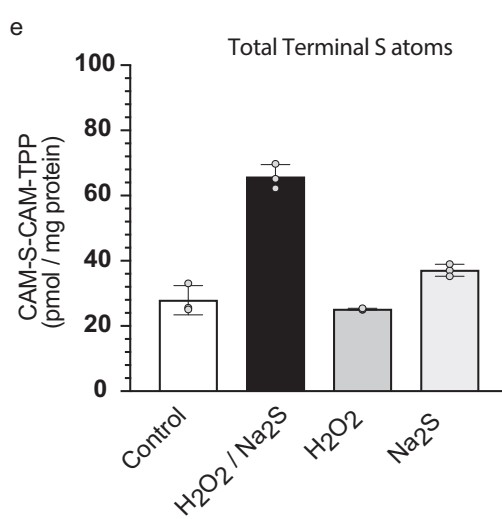

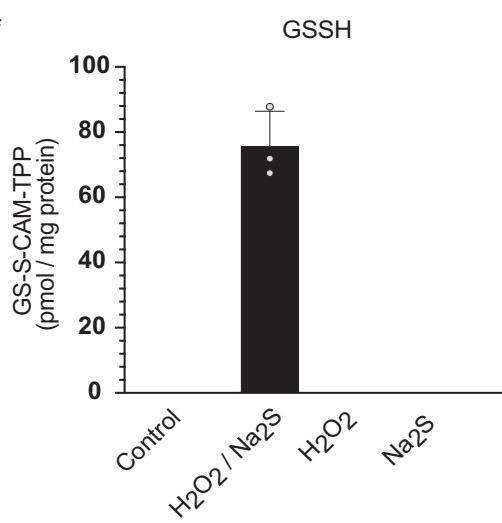

**Fig. 6 | Analysis of the ischaemic heart and oxidatively stressed mitochondria.**
**a**–**d** Mouse hearts were processed as in Fig. 5a. Hearts were rapidly processed following death (Normoxia) or incubated under ischaemic conditions (ischemia) and S(CAM-TPP)$_2$ (**a**) or TPP-CAM-S-CAM (**b**) quantified. Mouse hearts perfused on a Langendorff rig were processed after normoxia, normoxia + ischaemia, or after reperfusion following ischaemia and reperfusion, and S(CAM-TPP)$_2$ (**c**) or TPP-CAM-S-CAM (**d**) quantified. **e** Thiol reaction with $H_2O_2$ to disulfides/sulfenic acids that react with $H_2S$ to generate persulfides. **f** Mitochondria were treated with $Na_2S$ and/or $H_2O_2$, pelleted and processed to quantify CAM-S-CAM-TPP (**e**) or GSS-CAM-TPP (**f**). Data are means ± S.D. n = 4 (**a**, **b**) n= 5 (**c**, **d**), where n denotes the number of individual biological and n = 3 (**e**, **f**) where n denotes the number of technical replicates. Source data are provided as a Source Data file.

series of chemically stable standard molecules along with their deuterated internal standards enables absolute quantification of H₂S, RSS⁻, RS(S)ₙS⁻, and GSS⁻ in biological samples by construction of standard curves that account for tissue matrix effects. This suite of methods can now be applied to assess the role of persulfide and polysulfide formation in a wide range of biological situations.

We used this approach to quantify the levels of H₂S and of total, protein and low molecular weight terminal sulfur atoms on per- and polysulfides in a range of mouse and rat tissues. We found H₂S levels in the range 4–30 pmol/mg while the levels of terminal sulfur atoms due to persulfides and polysulfides were in the range 2–9 pmol/mg tissue. The dominant L$_{MW}$SS⁻ within tissues was GSS⁻, consistent with the far higher levels of GSH compared to other L$_{MW}$ thiols. Assuming ~70% tissue water by weight, these values correspond to tissue concentrations of H₂S of 6–40 μM, while the corresponding levels of terminal sulfur atoms on per-and polysulfides are 3–13 μM, all of which are within the range of previous reports[4,22,25,31,32].

To alter physiological H₂S levels, we exposed heart tissue to ischaemia, which is predicted to increase the levels of H₂S due to inhibition of its degradation by SQOR through the reduced CoQ pool[12,13]. This anticipated increase in H₂S was confirmed by trapping it as the diagnostic product S(CAM-TPP)₂. However, elevated H₂S was not associated with a change in the bulk levels of protein or low-molecular-weight persulfides. This suggests that elevation of H₂S alone is not sufficient to increase persulfide levels under these conditions. As H₂S is predicted to react with disulfides and sulfenic acids to form persulfides[16], we explored the effect of exposing mitochondria to H₂O₂ and H₂S simultaneously. This led to the formation of L$_{MW}$S(S)ₙS⁻, with GSS⁻ dominating this formation. This suggests that persulfides can form in response to oxidative stress in the presence of H₂S, which may act to decrease the irreversible oxidation of protein and L$_{MW}$ thiols[25]. Furthermore, as persulfides have a low pKa (~4.3) they are largely in the thiolate form in vivo, and due to the α-effect of the non-terminal S atom are better nucleophiles than thiols[29]. Thus, persulfides are far more effective antioxidants than thiols alone[4,22,54]. In summary, the beneficial effects of elevated H₂S under pathophysiological conditions, or following pharmacological intervention, may be due in large part to the antioxidant consequences of the generation of persulfides.

The quantitative approaches developed here are focused on bulk changes in the levels of H₂S and in the persulfidation status of thiols. The regulatory role of reversible protein persulfidation is therefore outside the scope of this study. However, our findings do suggest that the mechanism(s) by which proteins regulated by persulfidation incorporate a sulfur atom need to be carefully considered. Furthermore, the extent of modification of the putative regulatory thiol by persulfidation needs to be quantified, as low stoichiometries of modification can only be regulatory if they activate a gain-of-function of the modified protein. It is also likely that thiols most susceptible to persulfidation are also reversibly modified by a range of other processes, including glutathionylation, S-nitrosation and S-acylation, and whether these modifications are regulatory needs to be demonstrated in each case. One possibility that needs to borne in mind is that the majority of these protein thiol modifications may be a response to changes in the cellular milieu, rather than nodes of regulation[55].

In conclusion, exploration of the pathophysiological and pharmacological roles of H₂S and of persulfide/polysulfide formation in biological systems has been hampered by the lack of methods to quantify bulk changes. Here we have developed a robust and sensitive suite of methods that enables the sensitive and simultaneous quantification of H₂S, PrS(S)ₙS⁻ and L$_{MW}$S(S)ₙS⁻ in a range of biological samples, from isolated proteins to tissue homogenates. Here we used this approach to show that the steady-state levels of PrSS⁻ and L$_{MW}$SS⁻ within tissues are low and that GSS⁻ is the dominant L$_{MW}$SS⁻. Finally, we demonstrated that during ischaemia, H₂S levels rose due to the inactivation of SQOR, but that this elevation in H₂S did not increase

persulfides. In contrast, exposure to H₂O₂ in conjunction with H₂S led to extensive persulfide formation, consistent with a role for persulfides in antioxidant defence. The approaches we have introduced will facilitate progress in exploring the pathophysiological and pharmacological roles of H₂S in a wide range of biological scenarios.

## Methods

### Chemical syntheses

**Overview of synthesis.** The symmetrical disulfides, SS(CAM-TPP)₂ and SS(CAM-$d_{15}$TPP)₂ (where CAM = carbamidomethyl), and unsymmetrical sulfides, CAM-S-CAM-TPP and CAM-S-CAM-$d_{15}$TPP, were prepared from Dithiodiglycolic acid, SS(CM)₂ (where CM = carboxymethyl, Supplementary Fig. 2a). SS(CM)₂ was coupled with *N*-hydroxysuccinimide (NHS) to give diester SS(NHS-CM)₂. SS(NHS-CM)₂ was reacted with TPP-pentylamine or $d_{15}$TPP-pentylamine generated in situ from their hydrobromide salts to give the desired symmetrical disulfides, SS(CAM-TPP)₂ and SS(CAM-$d_{15}$TPP)₂, respectively. Reduction of the disulfide in the presence of iodoacetamide then led to the formation of the unsymmetrical sulfides, CAM-S-CAM-TPP and CAM-S-CAM-$d_{15}$TPP.

The symmetrical sulfides, S(CAM-TPP)₂ and S(CAM-$d_{15}$TPP)₂, were made from 2,2'-thiodiacetic acid, S(CM)₂, by coupling with TPP-pentylamine or $d_{15}$TPP-pentylamine generated in situ from their hydrobromide salts (Supplementary Fig. 2b). The glutathione sulfide products GS-CAM-TPP and GS-CAM-$d_{15}$TPP were prepared by direct displacement of iodide from IAM-TPP[40] and IAM-$d_{15}$TPP[40], respectively, with glutathione in the presence of base (Supplementary Fig. 2c). IAM-TPP and IAM-$d_{15}$TPP were reacted with thiosulfate to give the Bunte salts, O₃SS-CAM-TPP and O₃SS-CAM-$d_{15}$TPP, and the sulfite was displaced with glutathione to give the glutathione disulfide products, GSS-CAM-TPP and GSS-CAM-$d_{15}$TPP (Supplementary Fig. 2c).

### Dithiobis(succinimidyl acetate)[56,57] SS(NHS-CM)₂

*N,N'*-Dicyclohexylcarbodiimide (1.42 g, 6.87 mmol) was dissolved in anhydrous tetrahydrofuran (30 mL) under argon and cooled to 0 °C with stirring. Dithiodiglycolic acid [SS(CM)₂ 596 mg, 3.27 mmol] was added, followed by *N*-hydroxysuccinimide (790 mg, 6.87 mmol), and the solution was allowed to warm to RT over 2 h. The solution was diluted with ethyl acetate (20 mL), filtered through fluted filter paper and concentrated *in vacuo*. The crude material was purified by automated flash column chromatography on silica via gradient elution, from 100% dichloromethane to 1% methanol in dichloromethane to produce SS(NHS-CM)₂ (979 mg, 79%) as an off-white foamy solid. ¹H NMR (400 MHz, CDCl₃) δ 3.92 (s, 4H, H₂-1), 2.85 (s, 8H, H₂-2); ¹³C NMR (101 MHz, CDCl₃) δ 168.84 (C × 4), 165.23 (C × 2), 38.91 (CH₂ × 2), 25.76 (CH₂ × 4); *m/z* (ESI) 398.9927 (MNa⁺. C₁₂H₁₂N₂NaO₈S₂ requires 398.9927).

### SS(CAM-TPP)₂, dichloride salt

Triethylamine (201 μL, 1.45 mmol) was added dropwise to a stirred solution of SS(CAM-NHS)₂ (109 mg, 290 μmol) in anhydrous dichloromethane (3.0 mL) at (0 °C) stirring under argon. A solution of 5-aminopentyltriphenylphosphonium bromide hydrogenbromide[58] (368 mg, 724 μmol) in anhydrous dichloromethane (2.0 mL) was then

added dropwise to the solution and the mixture allowed to warm to RT over 3 h. The reaction was then cooled to 0 °C and quenched by addition of a saturated aqueous solution of ammonium chloride (3 mL). The solution was diluted with water (10 mL), extracted with dichloromethane (10 mL × 3), dried over MgSO$_4$, filtered and concentrated *in vacuo*. The crude material was purified by Isolera flash column chromatography on silica via gradient elution, from 100% dichloromethane to 20% methanol in dichloromethane to produce SS(CAM-TPP)$_2$ (92 mg, 32%) as a beige fluffy/foamy solid. $^1$H NMR (400 MHz, CDCl$_3$) δ 8.94 (t, *J* = 5.3 Hz, 2H, NH × 2), 7.88–7.62 (m, 30H, ArH × 30), 3.84–3.67 (m, 8H, H$_2$-1 × 2 and H$_2$-6 × 2), 3.07 (dd, *J* = 11.7, 5.8 Hz, 4H, H$_2$-2 × 2), 1.87–1.76 (m, 4H, H$_2$-4 × 2), 1.71–1.49 (m, 8H, H$_2$-3 × 2 and H$_2$-5 × 2); $^{13}$C NMR (126 MHz, CDCl$_3$) δ 168.71 (C × 2), 134.99 (d, *J* = 3.0 Hz, CH × 6), 133.91 (d, *J* = 10.0 Hz, CH × 12), 130.54 (d, *J* = 12.6 Hz, CH × 12), 118.46 (d, *J* = 85.8 Hz, C × 6), 41.18 (CH$_2$ × 2), 38.81 (CH$_2$ × 2), 27.96 (CH$_2$ × 2), 27.45 (d, *J* = 17.3 Hz, CH$_2$ × 2), 22.66 (d, *J* = 50.3 Hz, CH$_2$ × 2), 22.13 (d, *J* = 4.4 Hz, CH$_2$ × 2); $^{31}$P NMR (162 MHz, CDCl$_3$) δ 24.44; *m/z* (ESI) 421.1617 (M$^{2+}$. C$_{50}$H$_{56}$N$_2$O$_2$P$_2$S$_2$ requires 421.1624).

### SS(CAM-$d_{15}$TPP)$_2$, dichloride salt

5-Bromopentan-1-amine hydrobromide[59] (95 mg, 0.38 mmol) was dissolved in anhydrous acetonitrile (1.0 mL) deoxygenated by argon, and triphenylphosphine-$d_{15}$ (213 mg, 771 μmol) was added. The reaction vessel was then purged with argon and the mixture stirred and heated under reflux for 96 h. The reaction mixture was then cooled and concentrated *in vacuo*. The crude material was purified via trituration from ethyl acetate to give the corresponding triphenylphosphonium-$d_{15}$ product (240 mg) as a beige foamy solid, sufficiently pure for the next step. A solution of this sample (234 mg, 446 μmol) in anhydrous dichloromethane (2.0 mL) was added dropwise to a solution of SS(CAM-NHS)$_2$ (68.7 mg, 183 μmol) and triethylamine (124 μL, 893 μmol) in anhydrous dichloromethane (2 mL, prepared by dropwise addition of the triethylamine to the cooled solution of SS(CAM-NHS)$_2$) stirred under argon at 0 °C. The mixture was allowed to warm to RT over 24 h. The reaction was then cooled to 0 °C and quenched by addition of a saturated aqueous solution of ammonium chloride (2 mL). The solution was diluted with water (10 mL), extracted with dichloromethane (10 mL × 3), dried over MgSO$_4$, filtered and concentrated *in vacuo*. The crude material was purified by Isolera flash column chromatography on silica via gradient elution, from 100% dichloromethane to 20% methanol in dichloromethane to produce SS(CAM-$d_{15}$TPP)$_2$ (104 mg, 55% from SS(CAM-NHS)$_2$) as a beige fluffy/foamy solid. $^1$H NMR (400 MHz, CDCl$_3$) δ 8.89 (t, *J* = 5.3 Hz, 2H, NH × 2), 3.74–3.61 (m, 8H, H$_2$-1 × 2 and H$_2$-6 × 2), 3.07 (dd, *J* = 11.8, 5.9 Hz, 4H, H$_2$-2 × 2), 1.82–1.71 (m, 4H, H$_2$-4 × 2), 1.68–1.48 (m, 8H, H$_2$-3 × 2 and H$_2$-5 × 2); $^{13}$C NMR (101 MHz, CDCl$_3$) δ 168.68 (C × 2), 134.93–134.18 (m, CD × 6), 133.80–133.00 (m, CD × 12), 130.45–129.66 (m, CD × 12), 118.15 (d, *J* = 85.7 Hz, C × 6), 41.29 (CH$_2$ × 2), 38.75 (CH$_2$ × 2), 27.95 (CH$_2$ × 2), 27.38 (d, *J* = 17.1 Hz, CH$_2$ × 2), 22.64 (d, *J* = 50.0 Hz, CH$_2$ × 2), 22.04 (d, *J* = 4.4 Hz, CH$_2$ × 2); $^{31}$P NMR (162 MHz, CDCl$_3$) δ 24.18; *m/z* (ESI) 436.2569 (M$^{2+}$. C$_{50}$H$_{26}$D$_{30}$N$_2$O$_2$P$_2$S$_2$ requires 436.2565).

### CAM-S-CAM-TPP chloride

Iodoacetamide (68 mg, 0.37 mmol) was added to a stirred solution of SS(CAM-TPP)$_2$ (46 mg, 46 μmol) in anhydrous methanol (5.0 mL) at 0 °C. This was followed by addition of sodium borohydride (8.7 mg, 0.23 mmol) and the mixture was allowed to warm to RT over 4 h. The reaction mixture was then cooled to 0 °C and quenched by addition of a saturated aqueous solution of ammonium chloride (3 mL). The solution was diluted with water (10 mL), extracted with dichloromethane (10 mL × 3), dried over MgSO$_4$, filtered and concentrated *in vacuo*. The crude material was purified by Isolera flash column chromatography on silica via gradient elution, from 100% dichloromethane to 15% methanol in dichloromethane to produce CAM-S-CAM-TPP (9.1 mg, 16%) as a pale yellow fluffy/foamy solid. $^1$H NMR (500 MHz, CDCl$_3$) δ 7.88–7.67 (m, 16H, NH and ArH × 15), 7.61 (s, 1H, N*H*H), 5.88 (s, 1H, NH*H*), 3.62–3.53 (m, 2H, H$_2$-7), 3.46 (s, 2H), 3.40 (s, 2H), 3.24 (dd, *J* = 12.0, 6.0 Hz, 2H, H$_2$-3), 1.81–1.61 (m, 6H, H$_2$-4, H$_2$-5 and H$_2$-6); $^{13}$C NMR (126 MHz, CDCl$_3$) δ 172.40 (C), 170.08 (C), 135.34 (d, *J* = 3.0 Hz, CH × 3), 133.80 (d, *J* = 10.0 Hz, CH × 6), 130.75 (d, *J* = 12.5 Hz, CH × 6), 118.18 (d, *J* = 86.1 Hz, C × 3), 38.97 (CH$_2$), 36.89 (CH$_2$), 36.64 (CH$_2$), 27.82 (CH$_2$), 27.56 (d, *J* = 16.7 Hz, CH$_2$), 22.96 (d, *J* = 50.4 Hz, CH$_2$), 22.11 (d, *J* = 4.1 Hz, CH$_2$); $^{31}$P NMR (162 MHz, CDCl$_3$) δ 24.14; *m/z* (ESI) 479.1912 (M$^+$. C$_{27}$H$_{32}$N$_2$O$_2$PS requires 479.1917).

### CAM-S-CAM-$d_{15}$TPP chloride

Iodoacetamide (39.5 mg, 214 μmol) was added to a stirred solution of SS(CAM-$d_{15}$TPP)$_2$ (27.6 mg, 267 μmol) in anhydrous methanol (3.0 mL) at 0 °C under argon. This was followed by addition of sodium borohydride (5.0 mg, 0.13 mmol) and the mixture was allowed to warm to rt over 4 h. The reaction was then cooled to 0 °C and quenched by addition of a saturated aqueous solution of ammonium chloride (3 mL). The solution was diluted with water (10 mL), extracted with dichloromethane (10 mL × 3), dried over MgSO$_4$, filtered and concentrated *in vacuo*. The crude material was purified by Isolera flash column chromatography on silica via gradient elution, from 100% dichloromethane to 15% methanol in dichloromethane to produce CAM-S-CAM-$d_{15}$TPP (9.1 mg, 27%) as a pale yellow fluffy/foamy solid. $^1$H NMR (400 MHz, CDCl$_3$) δ 7.84 (t, *J* = 5.4 Hz, 1H, NH), 7.58 (s, 1H, N*H$_2$*), 6.63 (s, 1H, N*H$_2$*), 3.64–3.54 (m, 2H, H$_2$-7), 3.49 (s, 2H), 3.47 (s, 2H), 3.19 (dd, *J* = 11.4, 5.7 Hz, 2H, H$_2$-3), 1.81–1.57 (m, 6H, H$_2$-4, H$_2$-5 and H$_2$-6); $^{13}$C NMR (126 MHz, CDCl$_3$) δ 173.71 (C), 170.64 (C), 134.96–134.55 (m, CD × 3), 133.79–133.15 (m, CD × 6), 130.59–130.13 (m, CD × 6), 117.96 (d, *J* = 85.8 Hz, C × 3), 39.42 (CH$_2$), 36.88 (CH$_2$), 36.68 (CH$_2$), 27.95 (CH$_2$), 27.82 (d, *J* = 17.1 Hz, CH$_2$), 22.96 (d, *J* = 48.8 Hz, CH$_2$), 22.24 (d, *J* = 4.2 Hz, CH$_2$); $^{31}$P NMR (162 MHz, CDCl$_3$) δ 23.95; *m/z* (ESI) 494.2856 (M$^+$. C$_{27}$H$_{17}$D$_{15}$N$_2$O$_2$PS requires 494.2858).

### S(CAM-TPP)$_2$, dichloride salt

A solution of 5-aminopentyltriphenylphosphonium bromide hydrogen bromide[58] (200 mg, 0.4 mmol), 2,2′-thiodiacetic acid (S(CAM)$_2$ 30 mg, 0.2 mmol), HBTU (182 mg, 0.48 mmol) and DIPEA (0.14 mL, 0.8 mmol) in anhydrous DMF (2.0 mL) was stirred under argon for 1 h. The solvent was then removed under reduced pressure and diluted in dichloromethane (100 mL) and washed with LiCl 2% solution (3 × 50 mL). The organic fractions combined and dried over MgSO$_4$, filtered and concentrated under *vacuo*. The resulting residue was purified by flash chromatography SiO$_2$ Agela 12 g dichloromethane:methanol (100:0) increasing to (90:10). Ion exchange to chloride using ion exchange resin was carried out. The material was

triturated from chloroform and ether, the precipitate was collected by filtration yielding S(CAM-TPP)$_2$ (SW-114, 47 mg, 27%) as a colourless hydroscopic foam. $\delta_H$ (400 MHz, CDCl$_3$): 8.79 (t, 2H, $J$ = 5.4 Hz, NH), 7.78–7.63 (m, 30H, CH), 3.55–3.48 (m, 4H, CH$_2$), 3.28 (s, 4H, CH$_2$), 3.11 (q, 4H, $J$ = 6.0 Hz, CH$_2$), 1.63–1.53 (br m, 12H, CH$_2$); $\delta_C$ (101 MHz, CDCl$_3$): 170.176 (C × 2), 135.18 (d, $J$ = 2.8 Hz, CH × 6), 133.57 (d, $J$ = 9.8 Hz, CH × 12), 130.59 (d, $J$ = 12.5 Hz, CH × 12), 118.10 (d, $J$ = 85.9 Hz, C × 6), 38.58 (CH$_2$ × 2), 36.56 (CH$_2$ × 2), 27.97 (CH$_2$ × 2), 27.27 (d, $J$ = 16.5 Hz, CH$_2$ × 2), 22.50 (d, $J$ = 50.8 Hz, CH$_2$ × 2), 21.83 (d, $J$ = 4.2 Hz, CH$_2$ × 2); $\delta_P$ (67 MHz, CDCl$_3$): 23.861; $m/z$ (ESI) 405.1763 (M$^{2+}$. C$_{50}$H$_{56}$N$_2$O$_2$P$_2$S requires 405.1763).

### S(CAM-$d_{15}$TPP)$_2$, dichloride salt

A solution of 5-aminopentyltripentadeuterotriphenylphosphonium bromide hydrogen bromide (102 mg, 0.2 mmol, prepared as for SS(CAM-$d_{15}$TPP)$_2$ above), 2,2'-thiodiacetic (15 mg, 0.1 mmol), HBTU (90 mg, 0.22 mmol) and DIPEA (0.07 mL, 0.4 mmol) in anhydrous DMF (1.0 mL) was stirred under argon for 1 h. The solvent was then removed under reduced pressure and diluted in dichloromethane (50 mL) and washed with LiCl 2% solution (3 × 25 mL). The organic fractions combined and dried over MgSO$_4$, filtered and concentrated under *vacuo*. The resulting residue was purified by flash chromatography SiO$_2$ Agela 12 g dichloromethane:methanol (100:0) increasing to (90:10). Ion exchange to chloride using ion exchange resin was carried out. The material was triturated from chloroform and ether, the precipitate was collected by filtration yielding S(CAM-$d_{15}$TPP)$_2$ (20 mg, 22%) as a colourless hydroscopic foam. $\delta_H$ (400 MHz, CDCl$_3$): 8.79 (t, 2H $J$ = 5.2 Hz, NH), 3.60–3.53 (m, 4H, CH$_2$), 3.33 (s, 4H, CH$_2$), 3.15 (q, 4H, $J$ = 5.8 Hz, CH$_2$), 1.66–1.56 (m, 12H); $\delta_C$ (101 MHz, CDCl$_3$): 170.26 (C × 2), 134.71 (CH × 6), 133.30 (CH × 12), 130.08 (CH × 12), 118.00, (d, $J$ = 85.8 Hz, C × 6), 38.64 (CH$_2$ × 2), 36.75 (CH$_2$ × 2), 28.01 (CH$_2$ × 2), 27.31 (d, $J$ = 16.6 Hz, CH$_2$ × 2), 22.55 (d, $J$ = 50.5 Hz, CH$_2$ × 2), 21.88 (d, $J$ = 4.3 Hz, CH$_2$ × 2); $\delta_P$ (67 MHz, CDCl$_3$): 23.89; $m/z$ (ESI) 420.2699 (M$^{2+}$. C$_{50}$H$_{26}$D$_{30}$N$_2$O$_2$P$_2$S requires 420.2705).

**Semi-preparative HPLC conditions for glutathione conjugates.** Glutathione conjugates were purified using a Shimadzu Prominence semi-prep HPLC fitted with a Phenomenex Luna Omega 5 μm PS C18 100 Å, LC Column 250 × 10 mm. Solvent system–5.0 mL/min 23% MeCN:(0.1% TFA:H$_2$0) isocratic.

### GS-CAM-TPP, bis trifluoroacetate salt

Glutathione (26.0 mg, 85 μmol, 1.1 eq) was added to a solution of IAM-TPP iodide[40] (50.0 mg, 77 μmol, 1.0 eq) and triethylamine (35 mL, 0.255 mmol, 3.0 eq) in methanol (1.0 mL). The solution was stirred under argon at RT for 3 h and then concentrated under vacuum. The residue was dissolved in the 1 M HCl$_{(aq)}$ and the resulting solution was purified by semi-prep HPLC to give the GS-CAM-TPP as a white hygroscopic foam (58 mg, 80%). $\delta_H$ (400 MHz, MeOD): 7.94–7.85 (3 H, m, ArH), 7.85–7.72 (12 H, m, ArH), 4.62 (1 H, dd, $J$ = 9.1, 4.9 Hz, H-4), 4.05 (1H, t, $J$ = 6.4 Hz, H-1), 3.89 (1 H, d, $J$ = 17.7 Hz, H-6 $^{A \text{ or } B}$), 3.85 (1 H, d, $J$ = 17.7 Hz, H-6 $^{A \text{ or } B}$), 3.51–3.37 (2 H, m, PCH$_2$), 3.25 (1 H, d, $J$ = 14.7 Hz, H-7 $^{A \text{ or } B}$), 3.192 (1 H, d, $J$ = 14.7 Hz, H-7 $^{A \text{ or } B}$), 3.188 (2 H, d, $J$ = 6.5 Hz, H-8), 3.10 (1 H, dd, $J$ = 14.0, 4.9 Hz, H-5 $^{A \text{ or } B}$), 2.84 (1 H, dd, $J$ = 14.0, 9.1 Hz, H-5 $^{A \text{ or } B}$), 2.59 (2 H, t, $J$ = 7.1 Hz, H-3), 2.35–2.22 (1 H, m, H-2 $^{A \text{ or } B}$), 2.22–2.11 (1 H, m, H-2 $^{A \text{ or } B}$),1.76–1.65 (2 H, m, CH$_2$), 1.66–1.50 (4 H, m, CH$_2$). $\delta_C$ (101 MHz, MeOD): 174.48 (C), 172.98 (C), 172.59 (C), 172.27 (C), 171.52 (C), 162.11 (q, $J$ = 35.7 Hz, (C)), 136.28 (d, $J$ = 3.0 Hz, CH), 134.81 (d, $J$ = 10.0 Hz, CH), 131.54 (d, $J$ = 12.6 Hz, CH), 119.94 (d, $J$ = 86.3 Hz, C), 54.01 (CH), 53.51(CH), 41.80 (CH$_2$), 40.09 (CH$_2$), 36.19 (CH$_2$), 35.25 (CH$_2$), 32.36 (CH$_2$), 29.46 (CH$_2$), 28.71 (d, $J$ = 16.8 Hz, CH$_2$), 27.06 (CH$_2$), 23.16 (d, $J$ = 4.3 Hz, CH$_2$), 22.66 (d, $J$ = 51.3 Hz, CH$_2$). $\delta_F$ (377 MHz, MeOD): −77.04*. $\delta_P$ (162 MHz, MeOD): 23.73. m/z (ESI): Found: 695.2674. C$_{35}$H$_{44}$O$_7$N$_4$PS requires $M^+$, 695.2663.

### GS-CAM-$d_{15}$TPP, bis trifluoroacetate salt

Glutathione (26.0 mg, 85 μmol, 1.1 eq) was added to a solution of IAM-$d_{15}$TPP iodide[40] (51.0 mg, 0.077 mmol, 1.0 eq) and triethylamine (35 mL, 0.255 mmol, 3.0 eq) in methanol (1.0 mL). The solution was stirred under argon at RT for 3 h and then concentrated under vacuum. The residue was dissolved in the 1 M HCl$_{(aq)}$ and the resulting solution was purified by semi-prep HPLC to give GS-CAM-$d_{15}$TPP as a white hygroscopic foam (63 mg, 87%). $\delta_H$ (400 MHz, MeOD): 4.62 (1 H, dd, $J$ = 9.1, 4.9 Hz, H-4), 4.06 (1H, t, $J$ = 6.4 Hz, H-1), 3.89 (1 H, d, $J$ = 17.8 Hz, H-6 $^{A \text{ or } B}$), 3.85 (1 H, d, $J$ = 17.8 Hz, H-6 $^{A \text{ or } B}$), 3.48–3.36 (2 H, m, PCH$_2$), 3.26 (1 H, d, $J$ = 14.8 Hz, H-7 $^{A \text{ or } B}$), 3.20 (1 H, d, $J$ = 14.8 Hz, H-7 $^{A \text{ or } B}$), 3.19 (2 H, d, $J$ = 6.4 Hz, H-8), 3.10 (1 H, dd, $J$ 14.0, 4.9 Hz, H-5 $^{A \text{ or } B}$), 2.85 (1 H, dd, $J$ = 14.0, 9.1 Hz, H-5 $^{A \text{ or } B}$), 2.59 (2 H, t, $J$ = 7.1 Hz, H-3), 2.33–2.22 (1 H, m, H-2 $^{A \text{ or } B}$), 2.22–2.12 (1 H, m, H-2 $^{A \text{ or } B}$),1.76–1.65 (2 H, m, CH$_2$), 1.65–1.51 (4 H, m, CH$_2$). $\delta_C$ (101 MHz, MeOD): 174.48 (C), 172.98 (C), 172.60 (C), 172.27 (C), 171.50 (C), 162.11 (q, $J$ = 35.7 Hz, C), 135.77 (t, $J$ = 23.8 Hz, CD)), 134.37 (td, $J$ = 25.0, 9.9 Hz, CD), 131.03 (td, $J$ = 25.1, 12.5 Hz, CD), 119.69 (d, $J$ = 86.2 Hz, C), 116.02 (q, $J$ = 284.5 Hz, C) 54.03 (CH), 53.49 (CH), 41.81 (CH$_2$), 40.10 (CH$_2$), 36.20 (CH$_2$), 35.25 (CH$_2$), 32.35 (CH$_2$), 29.44 (CH$_2$), 28.70 (d, $J$ = 16.9 Hz, CH$_2$), 27.05 (CH$_2$), 23.13 (d, $J$ = 4.3 Hz, CH$_2$), 22.67 (d, $J$ = 51.3 Hz, CH$_2$). $\delta_F$ (377 MHz, MeOD): -77.16*. $\delta_P$ (162 MHz, MeOD): 23.59. m/z (ESI): Found: 710.3618. C$_{35}$H$_{29}$D$_{15}$O$_7$N$_4$PS requires $M^+$, 710.3604.

### O$_3$SS-CAM-TPP zwitterion

Sodium thiosulfate pentahydrate (77 mg, 0.31 mmol, 2.0 eq) was added to a solution of IAM-TPP iodide[58] (100 mg, 155 μmol, 1.0 eq) in MeCN:H$_2$0 (1:1). The solution was stirred overnight at RT then concentrated under vacuum. The residue was dissolved in dichloromethane and washed with water twice. The organic layer was dried over magnesium sulfate and concentrate under vacuum to give O$_3$SS-CAM-TPP as a colourless solid (47 mg, 60%). $\delta_H$ (400 MHz, CDCl$_3$): 7.97 (1 H, t, $J$ = 5.4 Hz, NH), 7.87–7.76 (3 H, m, ArH), 7.76–7.66 (12 H, m, ArH), 3.67 (2 H, s, SCH$_2$), 3.41–3.25 (2 H, m, PCH$_2$), 3.22 (2 H, apparent q, $J$ = 5.7 Hz, NCH$_2$), 1.77–1.69 (2 H, m, CH$_2$), 1.67–1.58 (2 H, m, CH$_2$), 1.58–1.49 (2 H, m, CH$_2$). $\delta_C$ (101 MHz, CDCl$_3$) 171.19 (C), 135.35 (d, $J$ = 3.1 Hz, CH), 133.60 (d, $J$ = 9.9 Hz, CH), 130.78 (d, $J$ = 12.5 Hz, CH), 118.17 (d, $J$ = 86.0. C), 38.77 (CH$_2$), 38.35 (CH$_2$), 27.50 (CH$_2$), 26.84 (d,

$J = 16.7$ Hz, CH$_2$), 22.37 (d, $J = 51.3$ Hz, CH$_2$), 21.77 (d, $J = 4.3$ Hz, CH$_2$). $\delta_P$ (162 MHz, CDCl$_3$): 23.81. MS unobtainable.

### O$_3$SS-CAM-$d_{15}$TPP zwitterion

Sodium thiosulfate pentahydrate (75 mg, 0.30 mmol, 2.0 eq) was added to a solution of IAM-$d_{15}$TPP iodide[58] (100 mg, 152 µmol, 1.0 eq) in MeCN:H$_2$O (1:1). The solution was stirred overnight at RT then concentrated under vacuum. The residue was dissolved in dichloromethane and washed with water twice. The organic layer was dried over magnesium sulfate and concentrate under vacuum to give the O$_3$SS-CAM-$d_{15}$TPP as a colourless solid (74 mg, 94%). $\delta_H$ (400 MHz, CDCl$_3$): 7.96 (1 H, t, $J = 5.6$ Hz, NH), 3.63 (2 H, s, SCH$_2$), 3.34–3.22 (2 H, m, PCH$_2$), 3.18 (2 H, apparent q, $J = 5.8$ Hz, NCH$_2$), 1.79–1.66 (2 H, m, CH$_2$), 1.66–1.56 (2 H, m, CH$_2$), 1.56–1.44 (2 H, m, CH$_2$). $\delta_C$ (101 MHz, CDCl$_3$) 171.09 (C), 134.80 (t, $J = 23.8$ Hz, CD), 133.12 (td, $J = 24.8$, 9.9 Hz, CD), 130.22 (td, $J = 24.7$, 12.5 Hz, CD), 117.87 (d, $J = 85.8$ C), 38.67 (CH$_2$), 38.37 (CH$_2$), 27.54 (CH$_2$), 26.86 (d, $J = 16.7$ Hz, CH$_2$), 22.31 (d, $J = 51.3$ Hz, CH$_2$), 21.72 (d, $J = 4.3$ Hz, CH$_2$). $\delta_P$ (162 MHz, CDCl$_3$): 23.61. MS unobtainable.

### GSS-CAM-TPP, bis trifluoroacetate salt

Glutathione (53.0 mg, 0.173 mmol, 1.1 eq) was added to a solution of O$_3$SS-CAM-TPP (79.0 mg, 0.157 mmol, 1.0 eq) and triethylamine (66 mL, 0.471 mmol, 3.0 eq) in methanol (1.5 mL). The solution was stirred under argon at R.T. for 3 h and then concentrated under vacuum. The residue was dissolved in the 1 M HCl$_{(aq)}$ (~2 mL) and extracted with dichloromethane (2 × 3 mL). The aqueous layer was then purified by semi-prep HPLC to give the GSS-CAM-TPP as a white hygroscopic foam (15 mg, 10%). $\delta_H$ (400 MHz, MeOD): 7.92–7.85 (3 H, m, ArH), 7.84–7.72 (12 H, m, ArH), 4.77 (1 H, dd, $J = 9.5$, 4.4 Hz, H-4), 4.04 (1 H, t, $J = 6.3$ Hz, H-1), 3.90 (1 H, d, $J = 17.7$ Hz, H-6 $^{A or B}$), 3.85 (1 H, d, $J = 17.7$ Hz, H-6 $^{A or B}$), 3.48–3.36 (4 H, m, PCH$_2$ + H-7), 3.32–3.25 (1 H, m, H-5 $^{A or B}$ partially obscured by residual peaks in NMR solvent), 3.24–3.15 (2 H, m, H-8), 2.95 (1 H, dd, $J = 13.9$, 9.5 Hz, H-5 $^{A or B}$), 2.58 (2 H, t, $J = 7.1$ Hz, H-3), 2.31–2.11 (2 H, m, H-2), 1.78–1.65 (2 H, m, CH$_2$), 1.65–1.52 (4 H, m, CH$_2$). $\delta_C$ (101 MHz, MeOD): 174.50 (C), 172.86 (C), 172.59 (C), 171.54 (C), 171.49 (C), 161.44 (q, $J = 37.1$ Hz, CF$_3$CO), 136.29 (d, $J = 3.1$ Hz, CH), 134.82 (d, $J = 9.9$ Hz, CH), 131.54 (d, $J = 12.5$ Hz, CH), 119.94 (d, $J = 86.3$, C), 53.79 (CH), 53.55 (CH), 43.42 (CH$_2$), 41.83 (CH$_2$), 40.75 (CH$_2$), 40.23 (CH$_2$), 32.39 (CH$_2$), 29.47 (CH$_2$), 28.74 (d, $J = 16.9$ Hz, CH$_2$), 27.05 (CH$_2$), 23.15 (d, $J = 4.4$ Hz, CH$_2$), 22.67 (d, $J = 51.3$ Hz, CH$_2$). $\delta_F$ (377 MHz, MeOD): -77.28 *. $\delta_P$ (162 MHz, MeOD): 23.73. m/z (ESI): Found: 727.2382. C$_{35}$H$_{44}$O$_7$N$_4$PS$_2$ requires $(M + H)^+$, 727.2384.

### GSS-CAM-$d_{15}$TPP, bis trifluoroacetate salt

Glutathione (46 mg, 0.147 mmol, 1.1 eq) was added to a solution of thiosulfonate (68.0 mg, 0.133 mmol, 1.0 eq) and triethylamine (56 mL,

0.40 mmol, 3.0 eq) in methanol (1.5 mL). The solution was stirred under argon at RT for 3 h and then concentrated under vacuum. The residue was dissolved in the 1 M HCl$_{(aq)}$ (2 mL) and extracted with dichloromethane (2 × 3 mL). The aqueous layer was then purified by semi-prep HPLC to give GSS-CAM-d$_{15}$TPP as a white hygroscopic foam (23.6 mg, 18%). $\delta_H$ (400 MHz, MeOD): 4.77 (1 H, dd, $J = 9.5$, 4.4 Hz, H-4), 4.04 (1 H, t, $J = 6.3$ Hz, H-1), 3.90 (1 H, d, $J = 17.7$ Hz, H-6 $^{A or B}$), 3.85 (1 H, d, $J = 17.7$ Hz, H-6 $^{A or B}$), 3.48–3.36 (4 H, m, PCH$_2$ + H-7), 3.32–3.25 (1 H, m, H-5 $^{A or B}$ partially obscured by residual peaks in NMR solvent), 3.24–3.15 (2 H, m, H-8), 2.95 (1 H, dd, $J = 13.9$, 9.5 Hz, H-5 $^{A or B}$), 2.58 (2 H, t, $J = 7.1$ Hz, H-3), 2.31–2.11 (2 H, m, H-2), 1.78–1.65 (2 H, m, CH$_2$), 1.65–1.52 (4 H, m, CH$_2$). $\delta_C$ (101 MHz, MeOD): 174.49 (C), 172.86 (C), 172.60 (C), 171.54 (C), 171.49 (C), 161.66 (q, $J = 36.3$ Hz, CF$_3$CO), 135.79 (t, $J = 24.4$ Hz, CH), 134.39 (td, $J = 25.0$, 9.9 Hz, CH), 131.04 (td, $J = 25.2$, 12.6 Hz, CH), 119.7o (d, $J = 86.2$ Hz, C), 53.79 (CH), 53.52 (CH), 43.42 (CH$_2$), 41.83 (CH$_2$), 40.74 (CH$_2$), 40.24 (CH$_2$), 32.37 (CH$_2$), 29.46 (CH$_2$), 28.74 (d, $J = 16.8$ Hz, CH), 27.03 (CH$_2$), 23.13 (d, $= J$ 4.3 Hz, CH), 22.67 (d, $J = 51.3$ Hz, CH). $\delta_F$ (377 MHz, MeOD): -77.16*. $\delta_P$ (162 MHz, MeOD): 23.58. m/z (ESI): Found: 742.3321. C$_{35}$H$_{29}$D$_{15}$O$_7$N$_4$PS$_2$ requires $(M + H)^+$, 742.3325.

*An additional peak in these $^{19}$F NMR spectra at approx. -76.8 ppm is assigned to residual trifluoroacetic acid. Given the hygroscopic nature of TPP-containing products of GSH, the different number of counterions possible depending on protonation state and the potential presence of small amounts of residual trifluoroacetic acid, the concentration of stock solutions were determined using the UV/Vis extinction coefficient in comparison with other TPP salts rather than from the mass weighed out.

All other chemicals were obtained from commercial vendors. All buffers used in this study were prepared using MiliQ water (18.2 MΩ·cm), treated with Chelex-100 resin.

**LC-MS/MS analysis of CAM-S-CAM-TPP and S(CAM-TPP)$_2$.** MS1 and product ion scans were performed by direct infusions into a Xevo TQ-S triple quadrupole mass spectrometer (Waters, UK). CAM-S-CAM-TPP, S(CAM-TPP)$_2$, and their deuterated internal standards (250 nM in 20% ACN, 0.1% FA) were infused at 1 µl/min and spectra recorded from 100–1000 m/z every 5 s for 1 min. For product ion scans compounds were infused at 2 µl/min and MS2 spectra were recorded from 100–500 m/z every 2 s for 1 min for the parent mass of interest. MS settings were: Capillary voltage 3.0 kV, Cone voltage 30 V, source offset 50 V, desolvation temperature 350 °C, source temperature 100 °C. The expected and detected masses of the selected compounds are presented in the Supplementary Table 1.

LC-MS/MS analyses of CAM-S-CAM-TPP, S(CAM-TPP)$_2$ and their deuterated internal standards were performed using a Xevo TQ-S triple quadrupole mass spectrometer (Waters, UK). Samples in 20% ACN, 0.1% FA were kept at 4 °C prior to analysis of 2–10 µL of pure compound mixtures or reacted extracts by the autosampler into a 15 µL flow-through needle. Separations were performed on an I-Class ACQUITY UPLC BEH C18 column (1 × 50 mm, 130 Å, 1.7 µm; Waters, UK) with a UPLC filter (0.2 µm; Waters, UK) at 40 °C using a ACQUITY UPLC I-Class system (Waters, UK). The mobile phases were MS solvent A (5% ACN, 0.1% FA) and B (90% ACN, 0.1% FA) at a flow rate of 0.2 ml/min with the gradient (the proportion of MS solvent B is given in%) represented in the Supplementary Table 2.

The eluate was directed to waste for the period 0-0.9 min and was analyzed by MS for the remaining 2.1 min of the UPLC gradient. Compounds were detected by multiple reaction monitoring (MRM) with electrospray ionization in positive ion mode using the following MS method settings: source spray voltage, 3.4 kV; cone voltage, 3 V; Ion source temperature, 150 °C; desolvation temperature, 200 °C. Nitrogen and argon were used as the curtain and the collision gases, respectively. MS/MS transitions used for quantification are presented in the Supplementary Table 3.

For standard curves, CAM-S-CAM-TPP and S(CAM-TPP)$_2$, were prepared in 20% ACN, 0.1% FA at final concentrations of 0.001 μM–25 μM supplemented with 50 nM of deuterated internal standards. An injection volume of 2 μL was used and a standard curve generated for all data points in the linear detection range, with higher concentrations deemed as saturating and excluded. For tissue extracts obtained from experiments on Langendorff hearts, the peak shapes and separation were affected by interfering contaminants, so to overcome this we modified the HPLC gradient as described in the Supplementary Table 4, keeping the rest of the parameters unchanged.

**LC-MS/MS analysis of GS-TPP and GSS-TPP.** MS1 and product ion scans were performed by direct infusions into a Xevo TQ-S triple quadrupole mass spectrometer (Waters, UK). GS-TPP and GSS-TPP, and their deuterated internal standards (100 nM in MeOH) were infused at 10 μL/min and spectra recorded from 100–1500 m/z every 5 s for 1 min. For product ion scans compounds were infused at 2 μL/min and MS2 spectra were recorded from 100–1500 m/z every 2 s for 1 min for the parent mass of interest. MS settings were: Capillary voltage 3.0 kV, Cone voltage 30 V, source offset 50 V, desolvation temperature 350 °C, source temperature 100 °C. The expected and detected masses of the selected compounds are presented in the Supplementary Table 5.

LC-MS/MS analyses of GS-CAM-TPP and GSS-CAM-TPP and their deuterated internal standards were performed using a Xevo TQ-S triple quadrupole mass spectrometer (Waters, UK). Samples in 50% of buffer (50 mM HEPES; pH 7.4) and 50% of (20% ACN, 0.1% FA) were kept at 8 °C prior to analysis of 2 μL of pure compound mixtures or reacted extracts by the autosampler into a 15 μL flow-through needle. Separations were performed on an I-Class ACQUITY UPLC BEH C18 column (1 × 50 mm, 130 Å, 1.7 μm; Waters, UK) with a UPLC filter (0.2 μm; Waters, UK) at 30 °C using a ACQUITY UPLC I-Class system (Waters, UK). The mobile phases were MS solvent A (5% ACN, 0.1% FA) and B (90% ACN, 0.1% FA) at a flow rate of 0.2 ml/min with the gradient (the proportion of MS solvent B is given in %) represented in the Supplementary Table 6.

Compounds were detected by multiple reaction monitoring (MRM) with electrospray ionization in positive ion mode using the following MS method settings: source spray voltage 2.60 kV; cone voltage 15 V; Ion source temperature 150 °C; desolvation temperature 200 °C. Nitrogen and argon were used as the curtain and the collision gases, respectively. MS/MS transitions used for quantification are presented in the Supplementary Table 7.

For standard curves, GS-CAM-TPP and GSS-CAM-TPP were prepared in 50% of buffer (50 mM HEPES; pH 7.4) and 50% of (20% ACN, 0.1% FA) at final concentrations of 20 nM to 1 μM and 0.5 μM to 10 μM for GS-CAM-TPP, 50 nM to 1 μM and 0.5 μM to 10 μM for GSS-CAM-TPP supplemented with 25 μM of deuterated internal standards GS-CAM-$d_{15}$TPP and 5 μM of deuterated internal standards GSS-CAM-$d_{15}$TPP. An injection volume of 2 μl was used and a standard curve generated for all data points in the linear detection range with higher concentrations deemed as saturating and excluded.

**Expression and purification of human recombinant Trx1.** The pQE-60 bacterial expression plasmid kindly supplied by Prof Tobias Dick, encoding human thioredoxin 1 (CSAAA)-SBP-His[36] was transformed into electrocompetent *Eschericia coli* BL21(DE3) cells (Thermo Scientific, Cat#EC0114). A single colony was used to inoculate 50 mL of LB medium supplemented with 50 μg/mL ampicillin. The pre-culture was incubated overnight at 37 °C with shaking at 225 rpm. The pre-culture was then used to inoculate 1 L of LB medium supplemented with 50 μg/mL ampicillin and grown at 37 °C until reaching an OD of 0.6. The cells were then incubated with 1 mM IPTG for 4 h at 37 °C. The cells were harvested by centrifugation (4000 g, 10 min, 4 °C). The bacterial pellet

was lysed using B-PER solution (ThermoFisher Scientific) supplemented with protease inhibitor cocktail tablets (cOmplete, EDTA-free), 10 mM imidazole and 0.5 mM DTT. The lysate was then centrifuged (16000 × g, 30 min, 4 °C) and the supernatant was collected. Ni-NTA agarose beads were added to the supernatant and incubated for 1 h at 4 °C. The beads were washed with 15 mL of wash buffer (50 mM Tris pH 7.4, 300 mM NaCl, 0.5 mM DTT, 20 mM imidazole) and then with 5 mL elution buffer (50 mM Tris pH 7.4, 300 mM NaCl, 0.5 mM DTT, 500 mM imidazole). The eluted protein sample was dialysed into 50 mM Tris pH 7.4 using a 10 kDa cut-off cassette. Purified protein was aliquoted at 17 mg/ml into 250 μl aliquots under Argon, and stored at −80 °C.

The purity of expressed Trx1 was determined using 10% SDS-PAGE analysis of samples (10 μg of sample/lane) in both reducing (with 50 mM of β-mercaptoethanol, DTT or TCEP) and non-reducing conditions (Supplementary Fig. 10). Additionally, the purity and integrity of Trx1 were confirmed by electroblotting of samples resolved by 10% PAGE using semi-dry transfer on to a nitrocellulose membrane (22 μm) and after blocking with 5% BSA in TBS for 30 min at RT, the membrane was immunoprobed with mouse monoclonal anti Trx1 antibody (Abcam Cat# ab16965, RRID:AB_443587; clone 3A1; dilution 1: 1000 in 1% BSA in TBS) overnight at 4 °C. After washing (3x with TBS), the membrane was exposed to goat anti mouse IRDye680RD highly cross-adsorbed IgG antibody (LICORbio Cat# 926-68070, RRID:AB_10956588; dilution 1: 15000) for 2 h at RT in Intercept TBS blocking buffer in dark. Subsequently, washed membrane (5x with TBS) was visualised using Odyssey DLx laser scanning imager (LI-COR) and image was analysed using Image Studio Lite software (LI-COR).

**Generation and quantification of persulfides and polysulfides on Trx1 by procedures A and B, respectively.** NeutrAvidin beads (5 mL slurry 5 mL; ThermoFisher Scientific #29202) were briefly centrifuged (1000 × g) and washed once with 8 mL 50 mM HEPES buffer (pH 7.5) and two times with 8 mL of the same buffer supplemented with 1 mM TCEP. The washed beads were mixed with 0.5 mL (10 fold dilution of initial concentration) purified human recombinant Trx1 protein (0.85 mg), diluted to 8 mL with 50 mM HEPES buffer (pH 7.5) and supplemented with 1 mM TCEP. Beads were incubated for 1 h at 4 °C under gentle rotation and then loaded on an empty gel filtration spin column (ThermoFisher Scientific; #69705) and washed with 80 mL argon-purged 50 mM HEPES buffer (pH 7.4). Beads containing bound and reduced Trx1 were resuspended in 6 mL 50 mM HEPES buffer (pH 7.5) and 500 μL fractions were loaded on sealable micro gel filtration columns (ThermoFisher Scientific, #69705). Persulfide and polysulfide formation was initiated by exposure to various concentrations of Na$_2$S$_2$ (freshly prepared from a 10 mM stock solution of Na$_2$S$_2$ in argon-purged/Chelex-100 treated ddH$_2$O) for 5 min. Some samples were also treated with 5 mM TCEP or were pretreated with 5 mM IAM. In an alternative protocol samples were then incubated with 10 equivalents of DTNB (from freshly prepared 100 mM DTNB in EtOH) for 10 min, washed and exposed to 1.2 equiv. of Na$_2$S for 5 min. For some samples ZnCl$_2$ (1 mM) was added at the same time as Na$_2$S. Samples were then washed x3 with cold argon-purged 50 mM HEPES buffer (pH 7.5) and immediately treated by either Procedure A with 5 mM IAM-TPP or Procedure B with 5 mM IAM for 30 min at RT protected from light. Samples were washed x3 with cold argon-purged 50 mM HEPES buffer (pH 7.5), resuspended in 500 μL washing buffer and exposed first to 1 mM TCEP, and after 3 min, 5 mM IAM was added and incubated for a further 20 min (Procedure A) or IAM-TPP (Procedure B) for 30 min at RT protected from light. Analytes were eluted from the gel filtration columns and 20 μL mixed with 980 μL of 20% ACN/0.1% FA and analysed by LC-MS. All samples were created and analysed in triplicate.

**Generation and quantification of persulfides on Cofilin1.** Purified full-length human recombinant Cofilin 1 protein expressed in

 

*Escherichia coli* (Abcam #62958) (11 µg/sample) was incubated with GSH (1 mM), MitoPerSulf (100 µM) or both in 50 mM HEPES buffer (pH 7.4) for 7 min at RT, followed by alkylation with IAM-TPP (20 mM) at RT for 30 min in the dark. Samples were then precipitated by ice cold $H_2O$/MeOH/$CHCl_3$ (v/v 4:4:1) washed and resuspended in 50 mM HEPES buffer (pH 7.4). Samples were then sequentially reduced with TCEP (5 mM) and alkylated with IAM (10 mM) for 30 min at RT in the dark. Samples were then precipitated with ice-cold MeOH, and supernatants were retained and evaporated overnight using a SpeedVac system at 37 °C. Dried samples were then resuspended in 100 µl 20% ACN/0.1% FA, diluted (1: 50) in the same solvent, and CAM-S-CAM-TPP quantified using LC-MS/MS analysis. All experiments were performed in triplicate.

**Cell culture and preparation of soluble protein extract.** Human embryonic kidney cells (HEK 293, ATCC) were obtained from ATCC (#CRL-1573; ATCC authenticated by STR profiling) and grown in DMEM medium (Gibco) supplemented with 10% FCS, 1% penicillin–streptomycin solution (Gibco) under humidified atmosphere of 95% air/5% $CO_2$ at 37 °C. Upon reaching 80% confluency, cells were washed with cold PBS and mechanically scraped into 1 mL of ice-cold Chelex-100-treated PBS containing 0.05% v/v benzonase (Sigma; Cat. #E1014). The cell suspension was sonicated using mechanical ultrasonicator (4 s intervals of 50 Hz repeated 10 times) followed by 3 cycles of freezing of cell lysate in liquid $N_2$ and thawing at room temperature. Finally, cell lysate was additionally homogenised by dispensing the lysate 10 times using a G26 needle and syringe. The cell lysate was additionally centrifuged at 16,000 × g for 10 min at 4 °C and the clear supernatant was used immediately. All procedures were performed on wet ice unless stated differently. Protein concentration was determined using the bicinchoninic acid assay (ThermoFisher Scientific) with BSA as a standard. The animal tissue lysates were prepared as described here by using 10–20 mg of snap-frozen animal tissue per sample.

**Generation and quantification of protein persulfides and polysulfides in cell lysates.** HEK 293 cell lysate (200 µg) protein/sample in 50 mM HEPES buffer (pH 7.4) was treated with $Na_2S$ (0–40 µM) or $Na_2S_2$ (0–600 µM) for 7 min in sealed Eppendorf tubes with minimal head space at RT. After treatment, samples were alkylated either by Procedure A with IAM-TPP (1 mM) or by Procedure B with IAM (1 mM) for 30 min at RT in the dark. Following alkylation, samples were mixed with a SP3 magnetic beads mix (50% of SP3 hydrophilic and 50% of SP3 hydrophobic beads) (Cativia Cat. #45152105050250 and Cat. #65152105050250), incubated for 15 min and then 100% EtOH was added (900 µL) to adsorb protein and the beads isolated by magnetic capture and then beads were washed three times with 80% EtOH. Washed beads were resuspended in 100 µL of 50 mM HEPES buffer (pH 7.4) and sequentially reduced with TCEP (200 µM) and re-alkylated either with IAM (400 µM) Procedure A or with IAM-TPP (400 µM) Procedure B for 30 min at RT in the dark. After alkylation, 900 µL of 100% ACN was added to the bead slurry to re-adsorb proteins onto the beads and to precipitate protein, and the beads were captured magnetically. Supernatant obtained after elution from bead-adsorbed proteins was diluted (1:1) with 20% ACN/0.1% FA and mixed with corresponding deuterated internal standards (25 nM) and analysed by LC-MS. All experiments were performed in triplicate.

**Generation and quantification of LMW persulfides and polysulfides using GSH.** GSH (10 µM) in 100 µL 50 mM HEPES buffer (pH 7.4) was exposed to increasing concentration (0–10 µM) of DTNB for 20 min and then reacted with $Na_2S$ or $Na_2S_2$ (0 - 30 µM) for 5 min. Samples were then alkylated with 60 µM IAM-TPP or IAM for 20 min at RT in the dark and then reduced with 120 µM TCEP for 3 min and additionally alkylated with 240 µM IAM or IAM-TPP for 20 min at RT in the dark. After final alkylation, all samples were diluted (1:1) with 20% ACN/0.1%

FA containing 50 nM of deuterated internal standards and analyzed by LC-MS. All experiments were performed in triplicate using a 96-well plate. Negative controls were prepared by preincubating 10 µM GSH with 10 µM CNBF to block the thiol group prior to exposure to DTNB and $H_2S$ or by treating the obtained GSSH/GS(S)$_n$H samples with excess TCEP (100 µM) prior to alkylation and LC-MS analysis.

**Monitoring the reaction of GSH with DTNB and $Na_2S$.** DTNB (20 µM) was incubated in 50 mM HEPES buffer (pH 7.4) in 1.5 mL quartz cuvette with a stirring bar, and baseline absorbance was obtained at 412 $_{nm}$ using an Olis 8453 Diode-array UV/Vis spectrometer. After obtaining the baseline, GSH (20 µM) was added and the reaction monitored until it had reached maximal absorbance. Then $Na_2S$ (20 µM) was injected and the reaction was monitored until reached the maximal absorbance. Upon completing the reaction 20 µM IAM was injected in the cuvette and the decrease of absorbance was monitored.

**Generation and quantification of GSH, GSSH, persulfides and polysulfides from reaction of GSSG and $Na_2S$.** GSSG (100 µM) in 500 µL of 50 mM HEPES buffer (pH 7.4) was reacted with $H_2S$ (100–300 µM) for 20 min at RT in 500 µL gas tight tubes. In order to assess the levels of GSH and GSSH, 100 µL or the reaction mixture was blocked with IAM-TPP (1 mM) for 25 min at RT protected from light. After incubation, blocked samples containing GSH and GSSH were diluted (1:1) with 20% ACN/0.1% FA containing 25 µM of GS-CAM-$d_{15}$TPP and 5 µM of GSS-CAM-$d_{15}$TPP as internal standards.

To simultaneously quantify the levels of persulfide and polysulfide sulfur, one aliquot of reaction mixture (100 µL) was blocked with IAM-TPP (1 mM) for procedure A and another aliquot of the same reaction (100 µL) was blocked with IAM (1 mM) for procedure B for 20 min at RT protected from light. After blocking samples were reduced with TCEP (2.5 mM) for 3 min and re-alkylate with IAM (5 mM) for procedure A and with IAM-TPP (5 mM) for procedure B for 20 min at RT protected from light. After alkylation, all samples were diluted (1:1) with 20% ACN/0.1% FA containing 50 nM of deuterated internal standards.

All samples were prepared in triplicate and analysed by LC-MS/MS.

**Generation and quantification of GSH, GSSH, persulfide and polysulfides from reaction of GSH and $Na_2S$.** GSH (10 µM) in 400 µL of 50 mM HEPES buffer (pH 7.4) was reacted with $Na_2S_2$ (10–30 µM) for 5 min at RT in a 2 mL capacity 96-well plate under argon sealed with parafilm. In order to assess the levels of GSH and GSSH, 100 µL or reaction mixture was blocked with IAM-TPP (250 µM) for 25 min at RT protected from light. After incubation, blocked samples containing GS-CAM-TPP and GSS-CAM-TPP were diluted (1:1) with 20% ACN/0.1% FA containing 25 µM of GS-CAM-$d_{15}$TPP and 5 µM of GSS-CAM-$d_{15}$TPP as internal standards.

To simultaneously quantify the levels of persulfide and polysulfide sulfur, one aliquot of reaction mixture (100 µL) was blocked with IAM-TPP (250 µM) for procedure A and another aliquot of the same reaction (100 µL) was blocked with IAM (250 µM) for procedure B for 20 min at RT, protected from light. After blocking samples were reduced with TCEP (500 µM) for 3 min and re-alkylate with IAM (1 mM) for procedure A and with IAM-TPP (1 mM) for procedure B for 20 min at RT, protected from light. After alkylation, all samples were diluted (1:1) with 20% ACN/0.1% FA containing 50 nM of deuterated internal standards. All samples were prepared in triplicate and analysed by LC-MS/MS.

**Animal experiments.** Animal procedures on mice and rats were carried out in accordance with the UK Animals (Scientific Procedures) Act of 1986 and the University of Cambridge Animal Welfare Policy. Procedures were reviewed by the University of Cambridge Animal Welfare Ethical Review Board and approved to be carried out under the Project License PP4344323.

**Generation and quantification of protein persulfides and polysulfides in rat heart lysates using Na₂S₂.** Female Wistar rats (obtained from Charles River Laboratories, UK) were housed in pathogen-free facilities with *ad libitum* access to standard chow and water and were used for experiments at 9–12 weeks of age.

Rat heart tissue homogenate prepared as described above (native tissue lysate) was assessed using the BCA assay for protein concentration. Samples containing 100 µg of heart lysate in 200 µL of 50 mM HEPES buffer (pH 7.4) were treated with either buffer (control), Na₂S (40 µM) or Na₂S₂ (40 µM) for 5 min at RT. After treatment, each sample (100 µL) was mixed with 40 µL of SP3 beads mix (20 µL of hydrophobic + 20 µL of hydrophilic beads), incubated for 15 min and then overlayed with 1 mL of absolute EtOH. Subsequently, SP3 beads containing adsorbed proteins were magnetically captured and the beads resuspended in 200 µL of 50 mM HEPES buffer (pH 7.4) containing 5 mM IAM-TPP for procedure A or IAM for procedure B were further incubated in the dark for 30 min. After incubation, 1 mL of absolute EtOH was added and SP3 beads were magnetically captured and washed three times with 80% EtOH, then resuspended in 200 µL 50 mM HEPES buffer (pH 7.4) containing 1 mM TCEP and incubated for 3 min at RT. Subsequently, samples were treated with 2 mM of IAM for procedure A or IAM-TPP for procedure B for 20 min in the dark at RT and analytes were eluted from magnetically captured beads with 1 mL 100% ACN. ACN effluents were diluted (1:1) with 20% ACN/0.1% FA and 50 nM of deuterated internal standards. Prepared samples were immediately analysed by LC-MS. All experiments were performed in triplicates. The remaining SP3 beads were overlayed with 10 µL of 20% SDS, sonicated for 20 min and resuspended in 190 µL of PBS and protein were eluted from SP3 beads using magnetic stand. Protein concentration and the total protein amount per each sample was determined using BCA assay.

**Quantification of H₂S, total terminal and LMW terminal sulfur from rodent tissues.** Female Wistar rats (Charles River, UK), aged 9–12 weeks, or male C57BL/6 mice aged 8–12 weeks (Charles River, UK) were killed by stunning and cervical dislocation. Heart, liver, kidney and brain tissues were rapidly removed and clamped frozen at liquid N₂ temperature and stored at −80 °C.

Snap frozen rat and mouse heart tissues were homogenised using Percellys tubes (approximately 10 mg tissue/300 µL 50 mM HEPES (pH 7.4) buffer containing 10 mM IAM-TPP two times for 30 s). After homogenisation, tubes were centrifuged for 5 min at 16,000 × *g* at 4 °C and the obtained supernatant was incubated for 20 min at RT in the dark. Each sample was then split into three fractions (100 µL) and treated as follows: For H₂S measurement, 100 µL sample was precipitated with 1 mL of absolute MeOH at −40 °C for 2 h, centrifuged (16,000 × *g*), and obtained supernatant was evaporated and the residue was resuspended and sonicated in 100 µL of 50 mM HEPES buffer (pH 7.4). For total persulfide measurement, 100 µL samples were reacted with 20 mM TCEP for 3 min and then mixed with 40 mM IAM and further incubated at RT in the dark for 30 min. Samples were then precipitated with 1 mL absolute MeOH at −40 °C for 2 h, centrifuged (16,000 × *g* at 0 °C), and the obtained supernatants was evaporated and the residue was resuspended and sonicated in 100 µL 50 mM HEPES buffer (pH 7.4). To assess the LMW persulfides, 100 µL samples were precipitated with 1 mL of absolute MeOH at −40 °C for 2 h, centrifuged (16,000 × *g* at 0 °C), and obtained supernatants were evaporated and crude residues were resuspended and sonicated in 100 µL 50 mM HEPES buffer (pH 7.4). Resuspended samples were then reacted with 20 mM TCEP for 3 min and then mixed with 40 mM IAM and further incubated at RT in the dark for 30 min. After incubation, samples were evaporated again, and residues were resuspended and sonicated in 100 µL 50 mM HEPES buffer (pH 7.4). All above samples were diluted (1:1) with 20% ACN/0.1% FA, containing 100 nM of deuterated internal standards and analysed by LC-MS/MS. All experiments

were performed in triplicates. Precipitated proteins from all samples were air dried, overlaid with 10 µL of 20% SDS and resuspended in 90 µL of PBS. Protein concentration and protein amount were determined using the BCA assay.

**Langendorff experiments.** C57Bl/6J male mice (n = 15, ~22 g body weight) were purchased from Charles River (UK). Animals were housed in individually ventilated cages and maintained under controlled temperature (22 ± 2 °C) on a 12:12 h light and dark cycle with access to standard chow diet (irradiated PicoLab mouse diet 5058, LabDiet. I-DIET-5R58-9KG-BG, IPS UK) and water *ad libitum*. Enrichment was provided in the form of tunnels and chew sticks. Mice were administered terminal anaesthesia via intraperitoneal pentobarbitone injection (140 mg/kg) followed by cervical dislocation. Beating hearts were rapidly excised, cannulated, and perfused in isovolumic Langendorff mode with Krebs–Henseleit (KH) buffer continuously gassed with 95% O₂/5% CO₂ (pH 7.4, 37 °C) containing (in mM): NaCl (116), KCl (4.7), MgSO₄.7H₂O (1.2), NaHCO₃ (25), CaCl₂ (1.4), and glucose (11)[60–62]. Perfused hearts were divided into three protocols (n = 3 /group) group 1: normoxic equilibration (22 min); group 2: normoxic equilibration (22 min) followed by 20 min of global normothermic (37 °C) ischemia; group3: normoxic equilibration (22 min) followed by 20 min of global normothermic (37 °C) ischemia and 2 min of reperfusion. At the end of the protocols, hearts were freeze-clamped by Wollenberger tongs and stored in −80 °C.

**Quantification of H₂S and total persulfide from mice heart exposed to Langendorff model of ischemia-reperfusion.** Snap frozen mouse heart obtained from Langendorff experiments was homogenised using Percellys tubes (approximately 10 mg tissue per 300 µL of 50 mM HEPES (pH 7.4) buffer containing 10 mM IAM-TPP) two times for 30 s. After homogenisation, tubes were centrifuged for 5 min at 16,000 × *g* at 4 °C, and obtained supernatant was incubated for 20 min at RT in the dark. Each sample was then split into two fraction (100 µL) and treated as follows:

For H₂S measurement, 100 µL of each sample was precipitated with 1 mL of absolute MeOH at −40 °C for 2 h, centrifuged (16,000 × *g* at 0 °C), and obtained supernatant was evaporated and the residue was resuspended and sonicated in 100 µL of 50 mM HEPES buffer (pH 7.4).

For total persulfide measurement, 100 µL samples were reacted with 20 mM TCEP for 3 min and then mixed with 40 mM IAM and further incubated at RT in the dark for 30 min. Samples were then precipitated with 1 mL absolute MeOH at −40 °C for 2 h, centrifuged (16,000 × *g*), and the obtained supernatants were evaporated, and the residue was resuspended and sonicated in 100 µL 50 mM HEPES buffer (pH 7.4). All above samples were diluted (1:1) with 20% ACN/0.1% FA, containing 100 nM of deuterated internal standards and analysed by LC-MS/MS. All experiments were performed in triplicate.

**Animal LAD and MCAO experiments.** For LAD and MCAO experiments, male C57BL/6J mice aged 8–12 weeks were ordered from Charles River Laboratories UK (Margate, UK) and were housed in individually ventilated cages and maintained under controlled temperature (22 ± 2 °C) on a 12 h light/dark cycle with *ad libitum* access to chow and water.

**In vivo left anterior descending (LAD) coronary artery occlusion model.** The LAD model was used to induce an acute MI in situ as described previously[1]. Briefly, mice were anaesthetised by an intraperitoneal injection of sodium pentobarbital (70 mg/kg). Once anaesthetised (confirmed by toe-pinch reflex), the mice were intubated endotracheally and ventilated with 3 cm H₂O positive end-expiratory pressure. Temperature was kept at 37 °C using a rectal thermometer-controlled heatpad (TCAT-2LV, Physitemp, USA). Ventilation frequency was maintained at 110 breaths/min, with tidal volume

between 125 and 150 μl. To occlude the LAD coronary artery, the hearts were exposed, and a suture was placed around the prominent branch of the LAD and passed through a small plastic tube. The suture was then tightened so that the tube was pressed against the LAD to occlude the vessel, initiating ischaemia. At 30 min of ischaemia, the mice were sacrificed by cervical dislocation, and the myocardial area at-risk (ischaemic) and non-risk (control) were quickly issected and snap-frozen in liquid nitrogen. The occlusion was maintained to prevent reperfusion of the at-risk area.

**Murine middle cerebral artery occlusion (MCAO) stroke model.** The MCAO model was used to induce an acute stroke in situ as described previously[2]. Briefly, mice were anaesthetised with isoflurane (3% induction, 1.5–2% maintenance) given at 1 L/min O2 using TEC 3 Isoflurane Vaporiser (98723N, +Mediquip LTD). A Doppler flowmetry probe (PeriFlux System 5000, Perimed, Sweden) was attached to the ipsilateral side of the skull to continuously monitor and record cerebral blood flow (CBF) in the MCA territory. Next, the left common carotid artery (CCA) and external carotid artery were exposed and permanently ligated. The left ICA was temporarily clamped and an incision made on the CCA to allow the insertion of a 6–0 suture with a silicone-coated tip (602223PK10RE, Doccol, USA). Then, the vascular clamp was removed and the suture carefully advanced towards the MCA. Occlusion of the MCA was confirmed when a sudden drop in CBF was observed on the Doppler, confirming MCAO and the start of ischaemia. Animals that fail to register >70% drop in CBF relative to baseline were excluded from the study. After 30 min of ischaemia, mice were sacrificed by cervical dislocation with the occlusion suture left in situ to prevent reperfusion. The ipsilateral and contralateral hemispheres were rapidly dissected and snap-frozen in isopentane and stored in –80 °C until further analysis.

**Isolation of rat heart mitochondria.** Upon cervical dislocation of 9–12 weeks old male Wistar rats (Charles River, UK), heart mitochondria (RHM) were prepared by homogenization of their heart tissue in STEB buffer (250 mM sucrose, 5 mM Tris-HCl and 1 mM EGTA, pH 7.4) supplemented with 0.1% fatty acid–free BSA. After mechanical homogenisation, mitochondria were isolated by differential centrifugation ($2 \times 2450 \times g$ for 5 min, $2 \times 9150 \times g$ for 10 min at 4 °C) as brown pellets. Isolated mitochondria were washed twice with basic STE buffer (without 0.1% fatty acid–free BSA), resuspended in basic STE buffer and protein concentration of mitochondrial proteins was determined by the bicinchoninic acid (BCA) assay using BSA as a standard.

**Quantification of H2S and GSSH from oxidatively stressed mitochondria.** Rat heart mitochondria (1 mg protein/mL) were suspended in KCl buffer (pH 7.2) containing 1 mM EGTA and treated with carrier (KCl buffer), Na2S (250 μM) or H2O2 (50 μM) for 3 min, or H2O2 (50 μM) for 3 min followed by and Na2S (250 μM) for 3 min at RT. After incubation, mitochondria were pelleted (1 min at $16,000 \times g$) and pellets were resuspended in 200 μL of 50 mM HEPES buffer (pH 7.4) supplemented with 10 mM IAM-TPP, disrupted using mechanical microhomogeniser and sonicated for 5 min. All samples were further incubated for 20 min at RT in the dark. After blocking, samples were centrifuged (1 min at $16,000 \times g$) and obtained supernatants were divided in two fractions (50 μL/fractions). One fraction for analysis of GSH-derived metabolites was immediately precipitated with MeOH and after evaporation, obtained residue was resuspended in 50 μL of 50 mM HEPES (pH 7.4), diluted with 20% ACN (containing 0.1% FA and mixture of internal standards) and immediately analysed by LC-MS/MS. The second fraction for analysis of total sulfane sulfur was treated with 20 mM TCEP for 2 min and subsequently reacted with 40 mM IAM at RT for 30 min in dark. After the treatment, samples were precipitated with MeOH and after evaporation, obtained residue was dissolved in 50 μL of 50 mM HEPES buffer, diluted with 20% ACN

(containing 0.1% FA and mixture of internal standards) and processed for LC-MS/MS analysis. GSH content of control mitochondria was $1.83 \pm 0.2$ nmol/mg protein (means ± SD, n = 3).

**Reporting summary**
Further information on research design is available in the Nature Portfolio Reporting Summary linked to this article.

## Data availability
Source data are provided with this paper. All data supporting the findings of the study are available from the corresponding author(s) upon request. The transformed NMR spectra and raw NMR data files, together with scanned HRMS spectra, for synthesised compounds are deposited at https://researchdata.gla.ac.uk/1879/ Raw LC-MS/MS data are deposited at https://doi.org/10.5061/dryad.7m0cfxq8k. Source data are provided with this paper.

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

## Acknowledgements

This work was supported by the Medical Research Council UK (MC_UU_00028/4) and by Wellcome Trust Investigator awards to M.P.M. (220257/Z/20/Z) and R.C.H. (220257/B/20/Z and 110158/Z/15/Z). J.Lj.M. was additionally supported by the Biotechnology and Biological Sciences Research Council (BLAST Pump-Priming Award, Grant No: BB/W01825X/1). D.A. acknowledges a Wellcome Trust Career Re-Entry Fellowship (221604/Z/20/Z). We are grateful to Prof Tobias Dick for the gift of the Trx1 plasmid. The authors would also like to thank Dr. Julien Pruden for his assistance with illustration software. For the purpose of open access, the author has applied a CC BY public copyright licence to any Author Accepted Manuscript version arising from this submission.

## Author contributions

J.Lj.M., R.C.H. and M.P.M. designed experiments and conceptualised the study. J.Lj.M. and N.B. performed the LC-MS method development and analysis of initial in vitro experiments. N.B. and C.S.Y. optimised MRM analysis. J.Lj.M. and C.S.Y. performed and analysed the samples used in this study. S.J. performed expression and protein purification. J.L. performed surgical intervention on mouse brain and heart. D.A. performed Langendorff experiments. T.A.P. performed the isolation of RHM. A.H.H., S.W. and S.T.C. synthesized compounds under direction of R.C.H. A.M.J. assisted with conceptualisation of the study. J.Lj.M. and M.P.M. wrote the manuscript with the assistance of all listed authors.

## Competing interests

The authors declare no competing interests.
