## [Transparent Peer Review file · Nature Communications]

Simultaneous and sensitive quantification of protein and low molecular weight persulfides, polysulfides and H₂S in biological samples

Corresponding Author: Professor Michael Murphy

Version 0:

Reviewer comments:

Reviewer #1

(Remarks to the Author)

In the manuscript by Miljkovic and colleagues, "Quantification of protein and low molecular weight persulfides in vivo" a new LC-MS/MS method to quantify persulfides and polysulfides in proteins and in glutathione is presented. The method is based on the use of an alkylating agent developed previously by the group, IAM-TPP, that has the advantage of giving a good signal in MS because of its intrinsically charged phosphonium.

The method is very robust, and the authors underwent the important feat of synthesizing all the deuterated standards to be able to reliably quantify the species of interest. In addition, an issue regarding the conversion of the persulfide derivative to a thioether, previously proposed in the literature, is adequately addressed.

Both the method and the obtained results are useful and relevant to the community interested in sulfur biology and biochemistry. The data are overall well presented and the manuscript is clearly written. Nevertheless, I have some concerns and comments that I hope will lead to improvements.

Major concerns

1. Detection of polysulfides in addition to persulfides.

Figure 1 and parts of the text are misleading in that it is said that persulfides were detected, while polysulfides were detected as well. Both Procedures A and B lead to TPP-CAM-S-CAM products that arise from the terminal sulfur in both persulfides and polysulfides. Although some parts of the text are clear, several changes should be introduced to address this. For example:

- In Fig. 1, the "Persulfide" boxes in procedure A and B could better be labeled "terminal sulfur of persulfides and polysulfides" or "persulfide plus polysulfide". Accordingly, the "Polysulfide" box may be more appropriately labeled "internal sulfane sulfur".
- In the final paragraph in Page 3, "Treatment with Na₂S₂ followed by analysis using procedure A led to the concentration-dependent and saturable generation of CAM-S-CAM-TPP, consistent with Na₂S₂ generating PrSS- on Trx1 (Fig. 3b)". Based on this only you cannot say it is PrSS-, because you cannot distinguish it from PrS(Sn)S-. Please rephrase, like "PrSS- and/or PrS(Sn)S-... ". The next sentence confirms the formation of polysulfides.
- Importantly, persulfides may only be accurately measured under conditions in which there is very low polysulfide formation (probably most biological conditions). This should be discussed at some point, at the start of results or at the start of the discussion section. In other words, the method is not specific for persulfides, but for terminal sulfane sulfur atoms. It could be discussed that in most physiological conditions the polysulfide concentration will be low, but that this needs to be quantified to draw conclusions from the method utilized in this work.

2. Levels of metabolites seem too high.

The amounts measured in Figs 5 and 6 appear quite high. I suggest revising the calculations. Also, compare with the literature. For example:

- In 5b, the amount of H₂S in mouse kidney is 78 pmol/mg tissue. Assuming 70% water, this number represents 111 uM! This is much higher than other estimates, please revise and compare with other published values. Could iron sulfur centers

be entering in the measurement? Could the mg be actually grams? Or mg protein instead of mg tissue?

- In 5c, persulfides in mouse liver are 100 pmol/mg tissue, or 143 μ M. Again, revise the calculations and compare with literature data.

- Revise calculations and compare with other literature values also for the other panels in Figures 5 and 6.

Minor comments

Abstract. There is no mention to polysulfides (see the related major comment).

Abstract, "exposure of mitochondria to H₂S in conjunction with oxidative damage". All the authors did was add hydrogen peroxide and that's fine, but then they should write "H₂S together with H₂O₂" or something like that.

Introduction, page 1. Consider mentioning intestinal bacteria as a source of H₂S.

Introduction, page 1. Consider mentioning thiosulfate sulfur transferase as one of the enzymes involved in catabolism.

Introduction, page 1. To our knowledge, sulfite levels are very low, consider mentioning thiosulfate and sulfate as end products.

Introduction, page 2, top paragraph. The in vivo relevance of the reaction between nitric oxide and H₂S is not clear to us. Neither is the relevance of the reaction between H₂S and nitrosothiols. Consider deleting.

Introduction, page 2, top paragraph, last sentence. Consider deleting "as either an antioxidant or a signal".

Introduction. Consider mentioning polysulfides in the introduction instead of the results section. Also, consider extending the definition to HS_nS⁻ and RSS_nSR in addition to RSS_nS⁻.

Results, page 3, and Fig. 1. Consider writing the structure of IAM-TPP, and possibly also IAM, in Fig. 1.

Figure 1. CAM-S-CAM in the top right of the scheme should have green instead of blue sulfur.

Throughout the text, the products with IAM or IAM-TPP are called adducts, but strictly speaking they are not adducts. I suggest just saying products.

Figure 2 and associated text in Page 3. Do the authors know what breaks generate the MRM transitions? If so, clarify.

SEM. In many figures the data are shown as mean \pm SEM. Shouldn't the standard deviation be shown instead of the SEM? Please correct.

Page 3, bottom. The sentence "and this was prevented by the presence of ZnCl₂ which sequesters H₂S preventing the formation of persulfides 32 (Fig. 3d, e)." does not go in this paragraph, but in the next because it refers to DTNB + H₂S treatment. Besides, e corresponds to cofilin, it should be only d.

Fig. 3, legend, third line, b-e should be replaced with b-d. since e is cofilin.

Page 4, second paragraph. Note that in the sentence "The bead-adsorbed proteins were then processed by procedure A to generate CAM-S-CAM-TPP (Fig. 3g) and by procedure B to generate S(CAM-TPP)₂ (Fig. 3h).", it should be Fig. 3f and Fig. 3g, there is no Fig. 3h. Sentences below mention Fig. 3h and Fig. 3i, but it appears they were integrated to Fig. 3g and h and do not exist independently in this manuscript. Correct and refer to the actual Figure.

In page 4, this sentence is missing some condition to explain the result: "When the cell lysate sample that had been labelled with IAM-TPP after Na₂S₂ treatment was eluted from the SP3 beads and immunoprobed with anti-TPP rabbit antiserum 36 the overall protein-bound TPP signal was weakest in samples treated with Na₂S₂ consistent with TPPIAM-based alkylation of PrSS⁻ and PrS(S)nS⁻ terminal sulfurs that are then lost upon reduction (Extended Data Figs. 3b), in agreement with Fig. 1." It implies there was a reduction step, but it is not mentioned explicitly in the text, or the Figure. Please explain.

In Page 5, first paragraph, when discussing Figure 4, it draws my attention the low yield of persulfide observed in DTNB/Na₂S treatment. It seems to me that the reaction was done with a large air headspace, or even in uncapped tubes, and lots of H₂S were lost by diffusion to the atmosphere. Methods said it was done in 96-well plates? Please discuss the low yields.

Figure 4, reaction between GSSG and H₂S. Why are the yields of thiol and persulfide so low comparing to the initial concentrations? Explain and compare with published studies of this reaction.

Figure 4, last panel. Why is only 12 μ M GS-CAM-TPP detected from 100 μ M GSH? Explain.

Figure 4, legend, second line from bottom. Maybe h is not correct? Samples from g?

Figure 4, legend, referring to Fig. 4 g, h. Please specify in the text the concentration of GSSG used. It is indicated for all other conditions except this one.

In Extended Data Fig. 1, a. SS-(CAM)₂, could it be named differently? Because it is not the same as the IAM derivative, that is mentioned in the text, but the IAA derivative. I would suggest changing it to SS-(CM)₂, the next to SS-(CM-NHS)₂, and the third product where the amide is formed, do leave as it is, and so the fourth.

In Extended Data Fig. 1, b. S-(CAM)₂, could it be named differently? Because it is not the same as the IAM derivative, that is mentioned in the text, but the IAA derivative. I would suggest changing it to S-(CM)₂, and the product where the amide is formed, leave as it is.

If you correct this in these figures, also correct the name in Methods / Chemical syntheses / Overview of Synthesis section.

In Page 5, second paragraph "This enabled the quantification of GSS- against internal deuterated standards (Extended Data Fig. 5b)." Maybe specify that you have synthesized GSS-CAM-TPP, because "internal standard" is broader than the same molecule deuterated, it could refer to a similar molecule.

In the text is not clear that you synthesized and purified all these molecules, you have to go to the supplemental info to find out. Besides, the synthesis of all the standards is a strength of the manuscript, it should be highlighted.

Extended Data Fig 5, legend. In the third line from bottom there is an "and" that does not make sense. I guess it should be "and TCEP".

Page 7 in Discussion, line 3 from bottom, says that IAM-TPP reacts rapidly. How rapidly? Has it been measured? Otherwise, consider deleting.

Reviewer #2

(Remarks to the Author)

In the manuscript by Miljkovic and colleagues, "Quantification of protein and low molecular weight persulfides in vivo" a new LC-MS/MS method to quantify persulfides and polysulfides in proteins and in glutathione is presented. The method is based on the use of an alkylating agent developed previously by the group, IAM-TPP, that has the advantage of giving a good signal in MS because of its intrinsically charged phosphonium.

The method is very robust, and the authors underwent the important feat of synthesizing all the deuterated standards to be able to reliably quantify the species of interest. In addition, an issue regarding the conversion of the persulfide derivative to a thioether, previously proposed in the literature, is adequately addressed.

Both the method and the obtained results are useful and relevant to the community interested in sulfur biology and biochemistry. The data are overall well presented and the manuscript is clearly written. Nevertheless, I have some concerns and comments that I hope will lead to improvements.

Major concerns

1. Detection of polysulfides in addition to persulfides.

Figure 1 and parts of the text are misleading in that it is said that persulfides were detected, while polysulfides were detected as well. Both Procedures A and B lead to TPP-CAM-S-CAM products that arise from the terminal sulfur in both persulfides and polysulfides. Although some parts of the text are clear, several changes should be introduced to address this. For example:

- In Fig. 1, the "Persulfide" boxes in procedure A and B could better be labeled "terminal sulfur of persulfides and polysulfides" or "persulfide plus polysulfide". Accordingly, the "Polysulfide" box may be more appropriately labeled "internal sulfane sulfur".

- In the final paragraph in Page 3, "Treatment with Na₂S₂ followed by analysis using procedure A led to the concentration-dependent and saturable generation of CAM-S-CAM-TPP, consistent with Na₂S₂ generating PrSS- on Trx1 (Fig. 3b)". Based on this only you cannot say it is PrSS-, because you cannot distinguish it from PrS(Sn)S-. Please rephrase, like "PrSS- and/or PrS(Sn)S-...". The next sentence confirms the formation of polysulfides.

- Importantly, persulfides may only be accurately measured under conditions in which there is very low polysulfide formation (probably most biological conditions). This should be discussed at some point, at the start of results or at the start of the discussion section. In other words, the method is not specific for persulfides, but for terminal sulfane sulfur atoms. It could be discussed that in most physiological conditions the polysulfide concentration will be low, but that this needs to be quantified to draw conclusions from the method utilized in this work.

2. Levels of metabolites seem too high.

The amounts measured in Figs 5 and 6 appear quite high. I suggest revising the calculations. Also, compare with the literature. For example:

- In 5b, the amount of H₂S in mouse kidney is 78 pmol/mg tissue. Assuming 70% water, this number represents 111 μM! This is much higher than other estimates, please revise and compare with other published values. Could iron sulfur centers be entering in the measurement? Could the mg be actually grams? Or mg protein instead of mg tissue?

- In 5c, persulfides in mouse liver are 100 pmol/mg tissue, or 143 μ M. Again, revise the calculations and compare with literature data.
- Revise calculations and compare with other literature values also for the other panels in Figures 5 and 6.

Minor comments

Abstract. There is no mention to polysulfides (see the related major comment).

Abstract, "exposure of mitochondria to H₂S in conjunction with oxidative damage". All the authors did was add hydrogen peroxide and that's fine, but then they should write "H₂S together with H₂O₂" or something like that.

Introduction, page 1. Consider mentioning intestinal bacteria as a source of H₂S.

Introduction, page 1. Consider mentioning thiosulfate sulfur transferase as one of the enzymes involved in catabolism.

Introduction, page 1. To our knowledge, sulfite levels are very low, consider mentioning thiosulfate and sulfate as end products.

Introduction, page 2, top paragraph. The in vivo relevance of the reaction between nitric oxide and H₂S is not clear to us. Neither is the relevance of the reaction between H₂S and nitrosothiols. Consider deleting.

Introduction, page 2, top paragraph, last sentence. Consider deleting "as either an antioxidant or a signal".

Introduction. Consider mentioning polysulfides in the introduction instead of the results section. Also, consider extending the definition to HS_nS⁻ and RSS_nSR in addition to RSS_nS⁻.

Results, page 3, and Fig. 1. Consider writing the structure of IAM-TPP, and possibly also IAM, in Fig. 1.

Figure 1. CAM-S-CAM in the top right of the scheme should have green instead of blue sulfur.

Throughout the text, the products with IAM or IAM-TPP are called adducts, but strictly speaking they are not adducts. I suggest just saying products.

Figure 2 and associated text in Page 3. Do the authors know what breaks generate the MRM transitions? If so, clarify.

SEM. In many figures the data are shown as mean +- SEM. Shouldn't the standard deviation be shown instead of the SEM? Please correct.

Page 3, bottom. The sentence "and this was prevented by the presence of ZnCl₂ which sequesters H₂S preventing the formation of persulfides 32 (Fig. 3d, e)." does not go in this paragraph, but in the next because it refers to DTNB + H₂S treatment. Besides, e corresponds to cofilin, it should be only d.

Fig. 3, legend, third line, b-e should be replaced with b-d. since e is cofilin.

Page 4, second paragraph. Note that in the sentence "The bead-adsorbed proteins were then processed by procedure A to generate CAM-S-CAM-TPP (Fig. 3g) and by procedure B to generate S(CAM-TPP)₂ (Fig. 3h).", it should be Fig. 3f and Fig. 3g, there is no Fig. 3h. Sentences below mention Fig. 3h and Fig. 3i, but it appears they were integrated to Fig. 3g and h and do not exist independently in this manuscript. Correct and refer to the actual Figure.

In page 4, this sentence is missing some condition to explain the result: "When the cell lysate sample that had been labelled with IAM-TPP after Na₂S₂ treatment was eluted from the SP3 beads and immunoprobed with anti-TPP rabbit antiserum 36 the overall protein-bound TPP signal was weakest in samples treated with Na₂S₂ consistent with TPPIAM-based alkylation of PrSS⁻ and PrS(S)_nS⁻ terminal sulfurs that are then lost upon reduction (Extended Data Figs. 3b), in agreement with Fig. 1." It implies there was a reduction step, but it is not mentioned explicitly in the text, or the Figure. Please explain.

In Page 5, first paragraph, when discussing Figure 4, it draws my attention the low yield of persulfide observed in DTNB/Na₂S treatment. It seems to me that the reaction was done with a large air headspace, or even in uncapped tubes, and lots of H₂S were lost by diffusion to the atmosphere. Methods said it was done in 96-well plates? Please discuss the low yields.

Figure 4, reaction between GSSG and H₂S. Why are the yields of thiol and persulfide so low comparing to the initial concentrations? Explain and compare with published studies of this reaction.

Figure 4, last panel. Why is only 12 μ M GS-CAM-TPP detected from 100 μ M GSH? Explain.

Figure 4, legend, second line from bottom. Maybe h is not correct? Samples from g?

Figure 4, legend, referring to Fig. 4 g, h. Please specify in the text the concentration of GSSG used. It is indicated for all other

conditions except this one.

In Extended Data Fig. 1, a. SS-(CAM)₂, could it be named differently? Because it is not the same as the IAM derivative, that is mentioned in the text, but the IAA derivative. I would suggest changing it to SS-(CM)₂, the next to SS-(CM-NHS)₂, and the third product where the amide is formed, do leave as it is, and so the fourth.

In Extended Data Fig. 1, b. S-(CAM)₂, could it be named differently? Because it is not the same as the IAM derivative, that is mentioned in the text, but the IAA derivative. I would suggest changing it to S-(CM)₂, and the product where the amide is formed, leave as it is.

If you correct this in these figures, also correct the name in Methods / Chemical syntheses / Overview of Synthesis section.

In Page 5, second paragraph "This enabled the quantification of GSS- against internal deuterated standards (Extended Data Fig. 5b)." Maybe specify that you have synthesized GSS-CAM-TPP, because "internal standard" is broader than the same molecule deuterated, it could refer to a similar molecule.

In the text is not clear that you synthesized and purified all these molecules, you have to go to the supplemental info to find out. Besides, the synthesis of all the standards is a strength of the manuscript, it should be highlighted.

Extended Data Fig 5, legend. In the third line from bottom there is an "and" that does not make sense. I guess it should be "and TCEP".

Page 7 in Discussion, line 3 from bottom, says that IAM-TPP reacts rapidly. How rapidly? Has it been measured? Otherwise, consider deleting.

Reviewer #3

(Remarks to the Author)

Miljkovic et al., report an interesting strategy that combines chemical trapping and LC-MS/MS measurement to simultaneously quantify LMW and protein persulfides. Using this strategy, they determined LMWSS- and PrSS- in a range of rodent tissues under basal conditions and during ischaemia. In general, this work is technically well done with proper experimental controls, particularly in terms of method development and validation of LC-MS/MS-based quantification. Also, the manuscript is well-written and inspiring to read. However, the paper does not fully recognize previous advances in this area, which suggests that the present report represents a more modest advance over the state of the art for measuring H₂S or protein persulfidation alone. Thus, due to the somewhat incremental advances, this work is not appropriate for publication in Nature Communications. It could be more suited for a technical journal focused on analytical methods or a redox biology journal, but only after the authors address major issues below.

Literature should be cited appropriately and a critical comparison made against the present method.

Fig.6 showed that the levels of H₂S, but not total persulfides, increased during ischaemia. It is necessary for the authors to further validate this finding by using orthogonal approaches for protein persulfidation detection. Moreover, if it is true, how can this result be interpreted? Indeed, the formation of protein persulfides may be regulated by the crosstalk between H₂S and H₂O₂. As shown in the same figure, the authors demonstrated that H₂O₂ could facilitate H₂S-induced persulfide formation, in accordance with many previous reports. To the best of my knowledge, however, the levels of total ROS typically increase during ischaemia. Thus, it is still confusing that ischaemia-induced H₂S did not increase the levels of protein persulfides in the meantime.

Minor issue:

The term "in vivo" used throughout the manuscript may be misleading.

Reviewer #4

(Remarks to the Author)

The advances presented in this manuscript appear to be incremental rather than significantly novel and as such it is not recommended for publication in Nature Communications. Previous studies have already reported approaches for persulfide detection using alkylating agents. The main difference in the present work seems to be the use of a triphenylphosphine-conjugated iodoacetamide. However, as currently presented, the manuscript does not convincingly demonstrate that this method represents a significant advancement over existing techniques.

Additionally, there are concerns regarding the clarity of Figure 1 and the description of the underlying methodology. It is unclear how specifically the authors can distinguish between persulfides and polysulfides during their analysis. For instance, in Procedure A, the authors claim specific persulfide detection; however, during the reduction step, while the reduction of RSS-CAM-TPP by TCEP followed by IAM trapping to generate TPP-CAM-S-CAM appears relatively straightforward, the presence of polysulfides could lead to the formation of multiple products, including TPP-CAM-S-CAM. These additional products are not discussed or represented in the schematic under Procedure A.

In addition, while the authors state that Procedure B is intended for polysulfide and persulfide detection, the steps involved

are not clear, particularly regarding the conversion of R-S-S-S-CAM species to TPP-CAM-S-CAM-TPP. Also, it is not clear why CAM-S-CAM would not be indicative of H₂S production along with TPP-CAM-S-CAM-TPP in Procedure A. Without a clearer stepwise description of the potential reactions and their contributions under Procedure A vs. Procedure B, it is difficult to assess the reliability and specificity of the approach.

Given these concerns, it is difficult to evaluate the significance of the remaining results.

Reviewer #5

(Remarks to the Author)

Version 1:

Reviewer comments:

Reviewer #1

(Remarks to the Author)

The comments raised previously were addressed satisfactorily. New experiments and more controls were performed. The manuscript is very much improved. However, on reading the revised version we detected some problems that need to be addressed.

The main problem has to do with the calculations of tissue levels. The revised numbers report H₂S levels of 4-30 pmol/mg tissue of H₂S. Assuming 70 % water, these numbers translate into 6-40 μ M. However, the authors calculate 6-40 nM. Similarly, the numbers for polysulfides (2-9 pmol/mg tissue) represent 3-13 μ M, not nM. In other words, pmol/mg tissue is the same as micromol/kg. Assuming a density of 1 kg/L, pmol/mg gives roughly micromolar. The numbers should be checked and the comparison with the literature should also be checked and rediscussed.

Fig. 4, panel i. The legend says they mixed GSH 10 μ M with 0, 100, 200 and 300 μ M of Na₂S₂. However, the figure (x axis) says that the ratios were 1-3. Correct the one that is wrong. Correct also the text (line 258).

Figure 4 and extended data 6. Revise concentrations (for example, GSH, Na₂S, DTNB) in the figures, legends and associated methods and text, there seem to be inconsistencies. Besides, consider including concentrations in the legends.

Extended Figure 6 A. It is surprising, even considering the unsealed plate, that so little H₂S was measured when it was one of the reagents (for example, less than 5 nM from 10 μ M sulfide, 10 μ M GSH and 0 DTNB). Please check the figure and the associated text (for example, "background H₂S" when it was a reagent, line 237). If H₂S cannot be measured in those conditions, consider deleting the results.

Discussion, line 366. The authors say that the sulfurs are stabilized by electrophilic reactions (I would say nucleophilic) to generate stable products rather than labile disulfide (no, disulfides are also formed with IAMTPP). Edit this part of the text.

Oxidation states are mentioned in the context of exploiting the chemical differences of terminal and internal sulfurs. I agree that the sulfurs can be distinguished, but since there is inconclusiveness in the oxidation states of these sulfur compounds, I suggest deleting the parts about oxidation states.

Minor observations:

In the abstract: Line 27, correct "quantifiedvsimultaneously"

Line 142, correct "complementary".

Line 226 "nitrobenzofurazane". Delete the last e.

Line 297, correct "biologicla"

Line 366, correct "stablise"

Line 60, in the context of biological systems, "the direct reaction of thiols with sulfur polymers systems" is unclear, consider revising.

Line 1344, author contributions, "Oroborous measurements" are not shown in this manuscript.

Reviewer #2

(Remarks to the Author)

Note that this review was done with another reviewer, so it is a duplicate of the other.

The comments raised previously were addressed satisfactorily. New experiments and more controls were performed. The manuscript is very much improved. However, on reading the revised version we detected some problems that need to be addressed.

The main problem has to do with the calculations of tissue levels. The revised numbers report H₂S levels of 4-30 pmol/mg tissue of H₂S. Assuming 70 % water, these numbers translate into 6-40 uM. However, the authors calculate 6-40 nM. Similarly, the numbers for polysulfides (2-9 pmol/mg tissue) represent 3-13 uM, not nM. In other words, pmol/mg tissue is the same as micromol/kg. Assuming a density of 1 kg/L, pmol/mg gives roughly micromolar. The numbers should be checked and the comparison with the literature should also be checked and rediscussed.

Fig. 4, panel i. The legend says they mixed GSH 10 uM with 0, 100, 200 and 300 uM of Na₂S₂. However, the figure (x axis) says that the ratios were 1-3. Correct the one that is wrong. Correct also the text (line 258).

Figure 4 and extended data 6. Revise concentrations (for example, GSH, Na₂S, DTNB) in the figures, legends and associated methods and text, there seem to be inconsistencies. Besides, consider including concentrations in the legends.

Extended Figure 6 A. It is surprising, even considering the unsealed plate, that so little H₂S was measured when it was one of the reagents (for example, less than 5 nM from 10 uM sulfide, 10 uM GSH and O DTNB). Please check the figure and the associated text (for example, "background H₂S" when it was a reagent, line 237). If H₂S cannot be measured in those conditions, consider deleting the results.

Discussion, line 366. The authors say that the sulfurs are stabilized by electrophilic reactions (I would say nucleophilic) to generate stable products rather than labile disulfide (no, disulfides are also formed with IAMTPP). Edit this part of the text.

Oxidation states are mentioned in the context of exploiting the chemical differences of terminal and internal sulfurs. I agree that the sulfurs can be distinguished, but since there is inconclusiveness in the oxidation states of these sulfur compounds, I suggest deleting the parts about oxidation states.

Minor observations:

In the abstract: Line 27, correct "quantifiedvsimultaneously"

Line 142, correct "complementary".

Line 226 "nitrobenzofurazane". Delete the last e.

Line 297, correct "biologicla"

Line 366, correct "stablise"

Line 60, in the context of biological systems, “the direct reaction of thiols with sulfur polymers systems” is unclear, consider revising.

Line 1344, author contributions, “Oroborous measurements” are not shown in this manuscript.

Reviewer #3

(Remarks to the Author)

The manuscript has been largely improved, and now I recommend publishing this paper as it is.

Reviewer #4

(Remarks to the Author)

Most all of the criticisms of the previous review have been addressed, and I believe the current version is now much more appropriate for publication. I have just a few minor comments for the authors to consider –

(1) On page 3, line 95 it is stated that previous methods do not distinguish between H₂S, RSS- and RS(S)nS-, but I believe previous trapping studies coupled with MS detection do in fact do this.

(2) In Extended Data Figure 1, the TCEP should be labeled as such, i.e., indicate that R = 2-carboxylethyl.

(3) In general, the resolution of the figures, especially chemical structures, is poor and should be improved before publication.

Response to Reviewers

Reviewers 1 and 2 (Identical report)

Comment

In the manuscript “Quantification of protein and low molecular weight persulfides in vivo” a new LC-MS/MS method to quantify persulfides and polysulfides in proteins and in glutathione is presented. The method is based on the use of an alkylating agent developed previously by the group, IAM-TPP, that has the advantage of giving a good signal in MS because of its intrinsically charged phosphonium.

The method is very robust, and the authors underwent the important feat of synthesizing all the deuterated standards to be able to reliably quantify the species of interest. In addition, an issue regarding the conversion of the persulfide derivative to a thioether, previously proposed in the literature, is adequately addressed.

Both the method and the obtained results are useful and relevant to the community interested in sulfur biology and biochemistry. The data are overall well-presented and the manuscript is clearly written. Nevertheless, I have some concerns and comments that I hope will lead to improvements.

Response

We are grateful to the reviewers for their kind comments. It was particularly gratifying that they noted the important technical breakthrough introduced by our use of an intrinsically charged phosphonium as well as our synthesis of all the deuterated standards.

Comment

1. Detection of polysulfides in addition to persulfides.

Figure 1 and parts of the text are misleading in that it is said that persulfides were detected, while polysulfides were detected as well. Both Procedures A and B lead to TPP-CAM-S-CAM products that arise from the terminal sulfur in both persulfides and polysulfides. Although some parts of the text are clear, several changes should be introduced to address this.

Response

We agree with the reviewer that our presentation was misleading in not clearly separating out the contribution from terminal and internal sulfur atoms. To be clear, we trap terminal sulfur atoms from per- and polysulfides, then we separately trap the internal sulfur atoms from polysulfides. The quantification of terminal perthiolate and perthiolate-like sulfur atoms (in polysulfides) together is more useful than a simple distinction between types of molecule because the terminal atoms share a (-1) oxidation state, and similar chemical and biological properties. The internal sulfane sulfurs of polysulfides are similar to each other and quite different chemically and biologically from the terminal sulfur atoms with a different (0) oxidation state, and no charge. Throughout the manuscript we have now made the distinction between the different sulfur detected both in terms of types of sulfur atoms quantified and species detected. We appreciate the feedback by the reviewers and have now further refined our text to make these aspects clearer to the reader. We note that our method also enables simultaneous trapping of H₂S which we also did not emphasise sufficiently in the original submission. We have now provided additional detail on this aspect and have changed the title to account for the extensive rewriting of the manuscript.

Comment

- In Fig. 1, the “Persulfide” boxes in procedure A and B could better be labeled “terminal sulfur of persulfides and polysulfides” or “persulfide plus polysulfide”. Accordingly, the “Polysulfide” box may be more appropriately labeled “internal sulfane sulfur”.

Response

We thank the reviewers for their suggestion and we have made the suggested changes in the figure and throughout the text to enhance clarity. We have also added a new Figure, Extended data Figure 1, to further explain the process.

Comment

- In the final paragraph in Page 3, “Treatment with Na₂S₂ followed by analysis using procedure A led to the concentration-dependent and saturable generation of CAM-S-CAM-TPP, consistent with Na₂S₂ generating PrSS- on Trx1 (Fig. 3b).”. Based on this only you cannot say it is PrSS-, because you cannot distinguish it from PrS(Sn)S-. Please rephrase, like “PrSS- and/or PrS(Sn)S-... “. The next sentence confirms the formation of polysulfides.

Response

The reviewers are correct and we have made the suggested change to the text.

Comment

Importantly, persulfides may only be accurately measured under conditions in which there is very low polysulfide formation (probably most biological conditions). This should be discussed at some point, at the start of results or at the start of the discussion section. In other words, the method is not specific for persulfides, but for terminal sulfane sulfur atoms. It could be discussed that in most physiological conditions the polysulfide concentration will be low, but that this needs to be quantified to draw conclusions from the method utilized in this work.

Response

We agree completely with the reviewers and have expanded on this important point in the rewritten Introduction, Results and Discussion sections.

Comment

2. Levels of metabolites seem too high.

The amounts measured in Figs 5 and 6 appear quite high. I suggest revising the calculations. Also, compare with the literature. In 5b, the amount of H₂S in mouse kidney is 78 pmol/mg tissue. Assuming 70% water, this number represents 111 uM! This is much higher than other estimates, please revise and compare with other published values. Could iron sulfur centers be entering in the measurement? Could the mg be actually grams? Or mg protein instead of mg tissue? In 5c, persulfides in mouse liver are 100 pmol/mg tissue, or 143 uM. Again, revise the calculations and compare with literature data. Revise calculations and compare with other literature values also for the other panels in Figures 5 and 6.

Response

We are very grateful to the reviewer for pointing this out. We have now gone back and revised nearly all of the data in Figs 5 and 6. This analysis uncovered an important point that we had not addressed. The issue was linked to the standard curves used to quantify CAM-S-CAM-TPP and S-(CAM-TPP)₂. For all experiments reported in the first submission we used standard curves as shown in Fig 2, which relied on the use of pure standard and internal

standard compounds. This approach is suitable for most of the data we present in which the experimental sample is immobilised on beads, or is precipitated, so that the TPP-containing reagent IAM-TPP, IAM and the TCEP were washed away from the sample before reduction to generate the required analytes for LC-MS/MS. However, in Figures 5 and 6 the analysis of the tissue samples was technically more complicated and it was not possible to separate the analytical sample from the introduced reagents such as IAM-TPP and TCEP. Thus, there was carry over of these reagents into the MS sample which we have now determined interfered with the LC-MS/MS analysis due to matrix effects. To overcome this limitation, we have now repeated the analyses using standard curves in the presence of the indicated reagents (technically, these are called "Matrix-matching calibration curves" (containing IAM-TPP, TCEP and IAM) (Extended Data Figure 8). We found that these measures affected the quantified amounts of the respective sulfur species and the new values have now been incorporated into an extensively revised Figure 5 and 6. Again we emphasise our gratitude to the reviewers for prompting us to address this error.

We fully agree with the reviewer about the importance of comparing our values with those in the literature. Therefore, we have now addressing this point in the Discussion section.

Minor comments

Comment

Abstract. There is no mention to polysulfides (see the related major comment).

Response

We thank the reviewer for this suggestion. We have now revised the abstract to explicitly include polysulfides and H₂S.

Comment

Abstract, "exposure of mitochondria to H₂S in conjunction with oxidative damage". All the authors did was add hydrogen peroxide and that's fine, but then they should write "H₂S together with H₂O₂" or something like that.

Response

We agree with the reviewer's point. This has been removed from the revised abstract and is discussed more accurately in the text.

Comment

Introduction, page 1. Consider mentioning intestinal bacteria as a source of H₂S.

Response

We appreciate this suggestion and have added a sentence in the Introduction discussing intestinal microbiota as a source of endogenous H₂S as well as some other pathways that contribute to endogenous H₂S production.

Comment

Introduction, page 1. Consider mentioning thiosulfate sulfur transferase as one of the enzymes involved in catabolism.

Response

We have now included thiosulfate sulfur transferase among the key enzymes involved in H₂S catabolism, as recommended.

Comment

Introduction, page 1. To our knowledge, sulfite levels are very low, consider mentioning thiosulfate and sulfate as end products.

Response

We thank the reviewers for pointing this out. We have clarified that thiosulfate and sulfate are the predominant end products.

Comment

Introduction, page 2, top paragraph. The in vivo relevance of the reaction between nitric oxide and H₂S is not clear to us. Neither is the relevance of the reaction between H₂S and nitrosothiols. Consider deleting.

Response

We have deleted these points.

Comment

Introduction, page 2, top paragraph, last sentence. Consider deleting “as either an antioxidant or a signal”.

Response

We have deleted this phrase.

Comment

Introduction. Consider mentioning polysulfides in the introduction instead of the results section. Also, consider extending the definition to HS_nS⁻ and RSS_nSR in addition to RSS_nS⁻.

Response

As suggested, we have now moved the discussion of polysulfides to the Introduction and extended the chemical definition to include RSS_nSR, and related species.

Comment

Results, page 3, and Fig. 1. Consider writing the structure of IAM-TPP, and possibly also IAM, in Fig. 1.

Response

The structures of both IAM-TPP and IAM have now been added to Figure 1.

Comment

Figure 1. CAM-S-CAM in the top right of the scheme should have green instead of blue sulfur.

Response

Thank you for picking this up! The sulfur atom in the CAM-S-CAM product has been corrected to green in Figure 1.

Comment

Throughout the text, the products with IAM or IAM-TPP are called adducts, but strictly speaking they are not adducts. I suggest just saying products.

Response

We thank the reviewer for this clarification. We have replaced the term “adducts” with “products” throughout the manuscript.

Comment

Figure 2 and associated text in Page 3. Do the authors know what breaks generate the MRM transitions? If so, clarify

Response

Yes, we have worked extensively on MS/MS analysis of TPP compounds and know their fragmentation patterns well. For CAM-S-CAM-TPP the structure of the diagnostic fragment of 420.23 (435.36 for the d₁₅ derivative) is now shown on Fig 2. For S(CAM-TPP)₂ the structure of the diagnostic fragment of 262.2 (277.4 for the d₁₅ derivative) is now shown on Fig 2.

Comment

SEM. In many figures the data are shown as mean +/- SEM. Shouldn't the standard deviation be shown instead of the SEM? Please correct.

Response

The figures have been revised to display standard deviation (SD).

Comment

Page 3, bottom. The sentence “and this was prevented by the presence of ZnCl₂ which sequesters H₂S preventing the formation of persulfides 32 (Fig. 3d, e).” does not go in this paragraph, but in the next because it refers to DTNB + H₂S treatment. Besides, e corresponds to cofilin, it should be only d.

Response

Thanks for spotting this error, which has now been corrected.

Comment

Fig. 3, legend, third line, b-e should be replaced with b-d. since e is cofilin.

Response

Thanks for spotting this error, which has now been corrected.

Comment

Page 4, second paragraph. Note that in the sentence “The bead-adsorbed proteins were then processed by procedure A to generate CAM-S-CAM-TPP (Fig. 3g) and by procedure B to generate S(CAM-TPP)₂ (Fig. 3h).”, it should be Fig. 3f and Fig.3g, there is no Fig. 3h. Sentences below mention Fig. 3h and Fig. 3i, but it appears they were integrated to Fig. 3g and h and do not exist independently in this manuscript. Correct and refer to the actual Figure.

Response

Thanks for spotting this error, which has now been corrected.

Comment

In page 4, this sentence is missing some condition to explain the result: “When the cell lysate sample that had been labelled with IAM-TPP after Na₂S₂ treatment was eluted from the SP3 beads and immunoprobed with anti-TPP rabbit antiserum the overall protein-bound TPP signal was weakest in samples treated with TCEP consistent with TPP-IAM-based alkylation of PrSS- and PrS(S)_nS- terminal sulfurs that are then lost upon reduction (Extended Data Figs. 3b), in agreement with Fig. 1.” It implies there was a reduction step, but it is not mentioned explicitly in the text, or the Figure. Please explain.

Response

We apologise for this omission. The revised text now includes a reduction step following IAM-TPP labelling, clarifying that the weak signal observed is due to the reductive loss of alkylated terminal sulfane sulfur species (PrSS⁻ and PrS(S)_nS⁻) from the proteins.

Comment

In Page 5, first paragraph, when discussing Figure 4, it draws my attention the low yield of persulfide observed in DTNB/Na₂S treatment. It seems to me that the reaction was done with a large air headspace, or even in uncapped tubes, and lots of H₂S were lost by diffusion to the atmosphere. Methods said it was done in 96-well plates? Please discuss the low yields.

Response

The reviewers' observation regarding the low yields observed in the DTNB/Na₂S reaction is correct. This does indeed arise due to carrying out the reaction in 96-well plates in a 100 μL assay volume with a large headspace and without any sealing. Thus, as the reviewers have correctly spotted, a significant fraction of the volatile H₂S will be lost due to diffusion H₂S. However, it is important to note that this does not impact on the significance of the result which was to demonstrate the synthesis of a persulfide by a targeted procedure. We have now altered the text to make this point clear and to discuss the low yield.

Comment

Figure 4, reaction between GSSG and H₂S. Why are the yields of thiol and persulfide so low comparing to the initial concentrations? Explain and compare with published studies of this reaction.

Response

The synthesis of glutathione persulfide (GSSH) from GSSG and H₂S has been previously characterized by Fakhoury et al. STAR Protocols 3, 101424 2022, who used relatively high concentrations of GSSG (50 mM) with a 5-fold excess of Na₂S (250 mM), with GSSH quantified via cold cyanolysis. In their study, GSSH yields ranged between 9–13 mM, corresponding to roughly 20% conversion. In contrast, our experimental design employed lower, somewhat more physiological concentrations of GSSG (100 μM) and a lower excess of H₂S (100 and 300 μM). Under these conditions, we detected approximately 4 and 9 μM GSSH (i.e., 4% and 9% of the starting GSSG), and 6 and 16 μM GSH (6% and 16%, respectively). This lower yield is consistent with the slower rate of this second order reaction at lower concentrations and ratios of reactants. Furthermore, our reactions were conducted open to the atmosphere, further enabling loss of H₂S. Thus, our findings are consistent with literature, and we have modified the text to explain this more clearly.

Comment

Figure 4, last panel. Why is only 12 uM GS-CAM-TPP detected from 100 uM GSH? Explain.

Response

We thank the reviewer for pointing out this issue and apologize for the oversight. The correct concentration of GSH used in the experiment was 10 μM , not 100 μM , as was correctly stated in the *Materials and Methods* section of the original manuscript. The discrepancy in the legend was due to a typographical error, which has now been corrected. We have also extended and repeated the measurements of GS-CAM-TPP in this experiment and the revised data are now included in an updated Fig 4i.

Comment

Figure 4, legend, second line from bottom. Maybe h is not correct? Samples from g?

Response

The legend has been rewritten to address this error.

Comment

Figure 4, legend, referring to Fig. 4 g, h. Please specify in the text the concentration of GSSG used. It is indicated for all other conditions except this one.

Response

Thank you for pointing this out. We have now specified the concentration of GSSG used in the experiments corresponding to Fig. 4g and 4h in the figure legend (100 μM).

Comment

In Extended Data Fig. 1, b, S-(CAM)₂, could it be named differently? Because it is not the same as the IAM derivative, that is mentioned in the text, but the IAA derivative. I would suggest changing it to S-(CM)₂, and the product where the amide is formed, leave as it is. 25. If you correct this in these figures, also correct the name in Methods / Chemical syntheses /Overview of Synthesis section.

Response

We agree with the Reviewer's comment and have made the changes to the text and to the figure and also renamed the NHS esters that are also not an IAM derivative.

Comment

In Page 5, second paragraph “This enabled the quantification of GSS- against internal deuterated standards (Extended Data Fig. 5b).” Maybe specify that you have synthesized GSS-CAM-TPP, because “internal standard” is broader than the same molecule deuterated, it could refer to a similar molecule.

Response

We have now clarified in the revised text that we synthesized the specific deuterated internal standard GSS-CAM-d₁₅-TPP for absolute quantification.

Comment

In the text is not clear that you synthesized and purified all these molecules, you have to go to

the supplemental info to find out. Besides, the synthesis of all the standards is a strength of the manuscript, it should be highlighted.

Response

We have now emphasized that we synthesized the specific deuterated internal standard to enable absolute quantification several places of the revised Results and Discussion sections.

Comment

Extended Data Fig 5, legend. In the third line from bottom there is an “and” that does not make sense. I guess it should be “and TCEP”.

Response

Thank you for spotting this inconsistency. We have corrected the phrasing in the legend to accurately read “and TCEP,” ensuring proper interpretation of the experimental steps.

Comment

Page 7 in Discussion, line 3 from bottom, says that IAM-TPP reacts rapidly. How rapidly? Has it been measured? Otherwise, consider deleting.

Response

As suggested, we have removed the statement regarding the rapidity of IAM-TPP reactivity from the Discussion section, since the exact reaction kinetics have not been formally quantified.

Reviewer #3

Comment

Authors report an interesting strategy that combines chemical trapping and LC-MS/MS measurement to simultaneously quantify LMW and protein persulfides. Using this strategy, they determined LMWSS- and PrSS- in a range of rodent tissues under basal conditions and during ischaemia. In general, this work is technically well done with proper experimental controls, particularly in terms of method development and validation of LC-MS/MS-based quantification. Also, the manuscript is well-written and inspiring to read.

Response

We are grateful that the reviewer recognises the detailed work that has gone into the development of this method.

Comment

However, the paper does not fully recognize previous advances in this area, which suggests that the present report represents a more modest advance over the state of the art for measuring H₂S or protein persulfidation alone. Thus, due to the somewhat incremental advances, this work is not appropriate for publication in Nature Communications. It could be more suited for a technical journal focused on analytical methods or a redox biology journal, but only after the authors address major issues below.

Response

While we do not agree with the reviewer on this point, we can see how this view has arisen as we did not emphasise the important developments that underlie our suite of new analytic approaches. We have now substantially rewritten the manuscript to emphasise why this

approach is more than an incremental change for the field. In particular, our approach is a significant advance on existing methodologies for 5 major reasons:

- 1 We can quantify the different types of sulfur atom in persulfides, polysulfides and H₂S simultaneously, by trapping.
- 2 We quantify both low molecular weight and total persulfides/polysulfides simultaneously.
- 3 In parallel with determination of general low molecular weight persulfides/polysulfides by sulfur atom trapping, we also determine the levels of glutathione persulfide, which is the dominant intracellular low molecular weight thiol. That these two orthogonal assays give consistent results both corroborates the two approaches, and also opens the way to assessing novel LMW persulfides.
- 4 The far greater sensitivity achieved by the introduction of an intrinsically charged phosphonium cation into the structure of the products that contain the cognate sulfur atoms makes detection by LC-MS/MS applicable to small samples.
- 5 Through novel synthetic chemistry we were able to generate a large number of standard molecules as well as deuterated internal standards of these. These developments make possible for the first time the identification and quantification of the products by LC-MS/MS in conjunction with standard curves obtained with the applicable biological material in order to account for matrix effects.

Comment

Literature should be cited appropriately and a critical comparison made against the present method.

Response

We agree that we neglected to adequately discuss previous method and compare them critically with the suite of methods introduced here. We have now done so in the Introduction, Results and Discussion, comparing and contrasting our approach with the most widely used methods in the literature.

Comment

Fig.6 showed that the levels of H₂S, but not total persulfides, increased during ischaemia. It is necessary for the authors to further validate this finding by using orthogonal approaches for protein persulfidation detection. Moreover, if it is true, how can this result be interpreted? Indeed, the formation of protein persulfides may be regulated by the crosstalk between H₂S and H₂O₂. As shown in the same figure, the authors demonstrated that H₂O₂ could facilitate H₂S-induced persulfide formation, in accordance with many previous reports. To the best of my knowledge, however, the levels of total ROS typically increase during ischaemia. Thus, it is still confusing that ischaemia-induced H₂S did not increase the levels of protein persulfides in the meantime.

Response

There are two separate aspects to this point. The first is that during cardiac ischaemia the lack of oxygen and thus activity of mitochondrial SQOR means that H₂S builds up as has been shown by us and others. This finding was confirmed by us using our new approach to assess H₂S levels *ex vivo* as is shown in Fig 6a,c. Our findings show that under these conditions of ischaemia the elevation of H₂S does not lead to any persulfide formation (Fig 6b,d). We have now extended these measurements to two other models of ischaemia, cardiac ischaemia in the mouse LAD model and brain ischaemia in the mouse MCAO model and these new data are

now shown in Extended Data Figure 8. Consistent with our findings in the Langendorff model, these data also show no increase in protein persulfidation during ischaemia. This finding is consistent with what we know about mechanisms of persulfidation as the conditions of ischaemia are highly reducing and also acidic. H₂S does not react directly with thiols but requires other modifications which do not occur during ischaemia.

The findings of no increase in persulfidation upon reperfusion are more intriguing. As the reviewer notes, we show clearly in isolated mitochondria that H₂S alone does not lead to the formation of per/polysulfides but that exposure to oxidative stress (in this case hydrogen peroxide) is necessary for the formation of persulfides. That we do not get persulfide formation in this circumstance, even though the generation of mitochondrial ROS upon reperfusion is well established, suggests to us that upon reperfusion the introduction of oxygen leads to the rapid consumption of H₂S. Thus, there is no H₂S around to react with any sulfenic acids generated. To assess this possibility would require extensive further experiments that we feel are beyond the scope of this paper. We have now rewritten this section to make this point.

Comment

Minor issue: The term “in vivo” used throughout the manuscript may be misleading.

Response

We have addressed this throughout the manuscript to clarify this point, replacing "in vivo" with "ex vivo" in most cases.

Reviewer #4

Comment

The advances presented in this manuscript appear to be incremental rather than significantly novel and as such it is not recommended for publication in Nature Communications. Previous studies have already reported approaches for persulfide detection using alkylating agents. The main difference in the present work seems to be the use of a triphenylphosphine-conjugated iodoacetamide. However, as currently presented, the manuscript does not convincingly demonstrate that this method represents a significant advancement over existing techniques.

Response

While we do not agree with the reviewer on this point, we can see how this view has arisen as we did not emphasise the important new developments that underlie our suite of new analytic approaches. We have now substantially rewritten the manuscript to emphasise why we believe that this approach is a step change for the field. In particular, our approach is a significant advance on existing methodologies for 5 major reasons:

- 1 We can quantify the different types of sulfur atom in persulfides, polysulfides and H₂S simultaneously, by trapping.
- 2 We quantify both low molecular weight and total persulfides/polysulfides simultaneously.
- 3 In parallel with determination of general low molecular weight persulfides/polysulfides by sulfur atom trapping, we also determine the levels of glutathione persulfide, which is the dominant intracellular low molecular weight thiol.

- That these two orthogonal assays give consistent results both corroborates the two approaches, and also opens the way to assessing novel LMW persulfides.
- 4 The far greater sensitivity achieved by the introduction of an intrinsically charged phosphonium cation into the structure of the products that contain the cognate sulfur atoms makes detection by LC-MS/MS applicable to small samples.
 - 5 Through novel synthetic chemistry we were able to generate a large number of standard molecules as well as deuterated internal standards of these. These developments make possible for the first time the identification and quantification of the products by LC-MS/MS in conjunction with standard curves obtained with the applicable biological material in order to account for matrix effects.

We have now extensively rewritten the entire manuscript so that these points are made far more explicit. In addition, we have now added sections comparing and contrasting our approach with the most widely used methods in the literature. We hope these extensive additions will address this referee's concerns about.

Comment

Additionally, there are concerns regarding the clarity of Figure 1 and the description of the underlying methodology. It is unclear how specifically the authors can distinguish between persulfides and polysulfides during their analysis. For instance, in Procedure A, the authors claim specific persulfide detection; however, during the reduction step, while the reduction of RSS-CAM-TPP by TCEP followed by IAM trapping to generate TPP-CAM-S-CAM appears relatively straightforward, the presence of polysulfides could lead to the formation of multiple products, including TPP-CAM-S-CAM. These additional products are not discussed or represented in the schematic under Procedure A. In addition, while the authors state that Procedure B is intended for polysulfide and persulfide detection, the steps involved are not clear, particularly regarding the conversion of R-S-S-S-CAM species to TPP-CAM-S-CAM-TPP. Also, it is not clear why CAM-S-CAM would not be indicative of H₂S production along with TPP-CAM-S-CAM-TPP in Procedure A. Without a clearer stepwise description of the potential reactions and their contributions under Procedure A vs. Procedure B, it is difficult to assess the reliability and specificity of the approach. Given these concerns, it is difficult to evaluate the significance of the remaining results.

Response

This point was also made by Reviewer 1/2 and we apologise that the distinction between the detection of persulfides/polysulfides and H₂S was not made clearly in the first submission. This has been addressed now by rewriting the manuscript and also by redesigning and relabelling the key explanatory figures. To address the specific points raised: The reviewer is correct that in Procedure A the terminal S atom of *both* persulfides and polysulfides will generate TPP-CAM-S-CAM. In conjunction with Procedure B to determine internal sulfur atoms we could potentially infer the amount of persulfides by assuming the main polysulfide is the trisulfide. However, quantification of terminal perthiolate and perthiolate-like sulfur atoms (in polysulfides) together is more useful than a simple distinction between types of molecule because the terminal atoms share a (-1) oxidation state, and similar chemical and biological properties. The internal sulfane sulfurs of polysulfides are similar to each other and quite different chemically and biologically from the terminal sulfur atoms with a different (0) oxidation state, and no charge. Regarding the conversion of R-S-S-S-CAM species to TPP-CAM-S-CAM-TPP, in procedure B, this will also trap all non-terminal S atoms as TPP-CAM-S-CAM-TPP. We have now included a new Figure, Extended Data Figure 1, to show this process. Finally, the reviewer is correct that the generation of CAM-S-CAM in Procedure

B is indicative of H₂S content. However, as this product is neutral the sensitivity of its detection by LC/MS-MS is far less than for S(CAM-TPP)₂. Thus, for this reason it was not explored further here. But this does highlight a major advantage of our approach.

Reviewer #1 & #2

Comment

The comments raised previously were addressed satisfactorily. New experiments and more controls were performed. The manuscript is very much improved. However, on reading the revised version we detected some problems that need to be addressed.

Response

All additional points have been addressed in full, and the manuscript has been further revised to enhance precision, and overall quality.

Comment

The main problem has to do with the calculations of tissue levels. The revised numbers report H₂S levels of 4-30 pmol/mg tissue of H₂S. Assuming 70 % water, these numbers translate into 6-40 μ M. However, the authors calculate 6-40 nM. Similarly, the numbers for polysulfides (2-9 pmol/mg tissue) represent 3-13 μ M, not nM. In other words, pmol/mg tissue is the same as micromol/kg. Assuming a density of 1 kg/L, pmol/mg gives roughly micromolar. The numbers should be checked and the comparison with the literature should also be checked and rediscussed.

Response

The units and tissue concentration calculations have been corrected accordingly. All relevant values now fall within the micromolar (μ M) range, and the corresponding text in the Results and Discussion has been updated to reflect these corrections and align with literature comparisons.

Comment

Fig. 4, panel i. The legend says they mixed GSH 10 μ M with 0, 100, 200 and 300 μ M of Na₂S₂. However, the figure (x axis) says that the ratios were 1-3. Correct the one that is wrong. Correct also the text (line 258).

Response

The concentrations have been re-checked and corrected to 0–30 μ M Na₂S₂, as now consistently indicated in the figure, legend, and corresponding text (line 258). This adjustment does not affect the data interpretation but ensures full accuracy and internal consistency.

Comment

Figure 4 and extended data 6. Revise concentrations (for example, GSH, Na₂S, DTNB) in the figures, legends and associated methods and text, there seem to be inconsistencies. Besides, consider including concentrations in the legends.

Response

All relevant concentrations have been harmonized across the text, figure legends, and Materials and Methods. Each reactant (e.g., GSH, Na₂S, DTNB) is now explicitly defined within the corresponding legends to provide full transparency and reproducibility.

Comment

Extended Figure 6 A. It is surprising, even considering the unsealed plate, that so little H₂S was measured when it was one of the reagents (for example, less than 5 nM from 10 μ M

sulfide, 10 μ M GSH and O DTNB). Please check the figure and the associated text (for example, “background H₂S” when it was a reagent, line 237). If H₂S cannot be measured in those conditions, consider deleting the results.

Response

The results referring to background H₂S measurements under these conditions have been removed to avoid confusion. The corresponding figure panel and related text have been revised to maintain clarity and accuracy.

Comment

Discussion, line 366. The authors say that the sulfurs are stabilized by electrophilic reactions (I would say nucleophilic) to generate stable products rather than labile disulfide (no, disulfides are also formed with IAMTPP). Edit this part of the text.

Response

The Discussion has been corrected to state that stabilization of reactive sulfur species occurs through nucleophilic reactions, generating stable products. The description now also acknowledges that asymmetric disulfides such as persulfides can be formed and subsequently blocked under the described conditions.

Comment

Oxidation states are mentioned in the context of exploiting the chemical differences of terminal and internal sulfurs. I agree that the sulfurs can be distinguished, but since there is inconclusiveness in the oxidation states of these sulfur compounds, I suggest deleting the parts about oxidation states.

Response

References to specific oxidation states have been removed to eliminate ambiguity. The text now focuses solely on distinguishing terminal and internal sulfur atoms based on their distinct chemical reactivity, without invoking uncertain oxidation state assignments.

Minor observations:

Comment

In the abstract: Line 27, correct “ quantifiedvsimultaneously”

Response

The abstract text has been verified, and this typographical error is no longer present. The sentence correctly reads quantified simultaneously.

Comment

Line 142, correct “complementary”.

Response

The word has been corrected to complementary.

Comment

Line 226 “nitrobenzofurazane”. Delete the last e.

Response

The term has been corrected to nitrobenzofurazan.

Comment

Line 297, correct “biologicla”

Response

The word has been corrected to biological.

Comment

Line 366, correct “stablise”

Response

The word has been corrected to stabilise.

Comment

Line 60, in the context of biological systems, “the direct reaction of thiols with sulfur polymers systems” is unclear, consider revising.

Response

This phrase has been removed and the sentence rewritten to improve clarity and accuracy.

Comment

Line 1344, author contributions, “Oroborous measurements” are not shown in this manuscript.

Response

Line 1344, author contributions, “Oroborous measurements” are not shown in this manuscript.

Reviewer #3 (Remarks to the Author):

Comment

The manuscript has been largely improved, and now I recommend publishing this paper as it is.

Response

No further revisions were required for this comment; the final version reflects all previously incorporated improvements.

Reviewer #4 (Remarks to the Author):

Comment

Most all of the criticisms of the previous review have been addressed, and I believe the

current version is now much more appropriate for publication. I have just a few minor comments for the authors to consider.

Response

All remaining minor points have been addressed as detailed below.

Comment

On page 3, line 95 it is stated that previous methods do not distinguish between H₂S, RSS- and RS(S)_nS-, but I believe previous trapping studies coupled with MS detection do in fact do this.

Response

The text has been revised to acknowledge that previous trapping studies coupled with MS detection are capable of distinguishing these species.

Comment

(2) In Extended Data Figure 1, the TCEP should be labelled as such, i.e., indicate that R = 2-carboxylethyl.

Response

The labels in Extended Data Figure 1 have been corrected to indicate explicitly that the R group corresponds to 2-carboxylethyl.

Comment

(3) In general, the resolution of the figures, especially chemical structures, is poor and should be improved before publication.

Comment

All figures, including chemical structures, have been replaced with high-resolution versions suitable for final publication.